# Synaptic input sequence discrimination on behavioral timescales mediated by reaction-diffusion chemistry in dendrites

**Upinder Singh Bhalla***

National Centre for Biological Sciences, Tata Institute of Fundamental Research, Bangalore, India

**Abstract** Sequences of events are ubiquitous in sensory, motor, and cognitive function. Key computational operations, including pattern recognition, event prediction, and plasticity, involve neural discrimination of spatio-temporal sequences. Here, we show that synaptically-driven reaction-diffusion pathways on dendrites can perform sequence discrimination on behaviorally relevant time-scales. We used abstract signaling models to show that selectivity arises when inputs at successive locations are aligned with, and amplified by, propagating chemical waves triggered by previous inputs. We incorporated biological detail using sequential synaptic input onto spines in morphologically, electrically, and chemically detailed pyramidal neuronal models based on rat data. Again, sequences were recognized, and local channel modulation downstream of putative sequence-triggered signaling could elicit changes in neuronal firing. We predict that dendritic sequence-recognition zones occupy 5 to 30 microns and recognize time-intervals of 0.2 to 5 s. We suggest that this mechanism provides highly parallel and selective neural computation in a functionally important time range.

**\*For correspondence:** bhalla@ncbs.res.in

## Introduction

Activity sequences have long been recognized as a fundamental constituent of neural processing. Lorente de No suggested that reverberatory activity sequences in small networks could sustain activity (*Lorente de No, 1938*). Hebb's idea of cell assemblies suggested that ensembles of cells encoded a particular neuronal concept, but also that there was sequential activation within the group of cells forming the assembly (*Hebb, 1949*).

Many sensory systems process sequential stimuli, and these are typically mapped to ensembles of sequentially active neurons (*Bouchard and Brainard, 2016*; *Broome et al., 2006*; *Carrillo-Reid et al., 2015*). Deeper in the brain, hippocampal place cells represent a higher-order cognitive map of space, yet here too sequences occur when the animal moves through spatial locations represented by the place cells (*Wilson and McNaughton, 1994*). The hippocampus exhibits other forms of sequential activity in the form of fast replay events (*Jadhav et al., 2012*; *Wilson and McNaughton, 1994*) and stimulus-bridging activity that emerges during associative learning (*MacDonald et al., 2011*) and trace conditioning (*Modi et al., 2014*). Thus, there are numerous neural correlates both of sequences, and of processing steps that recognize them.

The idea of synfire chains examines conditions for self-sustaining sequential activity to occur in multiple layers of a network (*Abeles, 1982*). This is non-trivial, as excessive activation can lead to runaway epileptiform activity, whereas insufficient activation causes decay of the activity wave (*Kumar et al., 2008*; *Mehring et al., 2003*). Neural networks that can recognize such sequences are well-established. For example, events may be run through a delay line, so that the oldest event is delayed more, the next event less, and so on, so that they all arrive at the neural network at the

same time. Thus, the time-sequence is flattened in time and classical attractor networks can recognize patterns in the sequence (*Tank and Hopfield, 1987*). Time-varying attractor networks have also been shown to be able to implement and recognize sequences (*Lee, 2002*). Time-invariance is a desirable feature of sequence recognition circuits, since the same order of events may take place at different speeds. Recursive networks using supervised learning, and short-term synaptic plasticity have been shown to be implement time-invariance (*Barak and Tsodyks, 2006*; *Goudar and Buonomano, 2015*; *Laje and Buonomano, 2013*).

While networks can carry out sequence recognition, they make limited use of the rich dynamics of biological neurons. The theory of Hierarchical Temporal Memory builds on the idea of sequence-recognizing neurons and networks as a way to perform complex temporal computation (*George and Hawkins, 2009*; *Hawkins and Ahmad, 2016*). Even the ability to recognize simultaneous closely-localized synaptic input has interesting computational implications (*Hawkins and Ahmad, 2016*), The current study extends this to patterns of input both in time and space.

One of the first biophysical proposals for subcellular sequence recognition was made by Rall, who showed theoretically that synaptic events propagate down the dendritic tree with a small delay. By timing synaptic inputs to coincide with this delay, Rall predicted that ordered input should yield a larger response than reversed input (*Rall, 1964*). A stronger version of this mechanism was experimentally demonstrated by *Branco et al. (2010)* who used glutamate uncaging to provide sequential input along a pyramidal neuron dendrite. They showed that NMDA receptor amplification of 'inward' sequences (distal to proximal) gave rise to about 40% larger somatic depolarization than 'outward' sequences, for rapid (~40 ms) sequences. However, many interesting neural sequences occur at slower time-scales.

At least three attributes should converge for single neurons to achieve and report sequence recognition in the noisy context of neural activity. First, the neurons should recognize inputs coming in the correct order in space and time. Second, the selectivity for the correct input over scrambled input and background noise should be strong enough for there to be reliable discrimination. Third, recognized sequences should trigger activity changes either by way of changed firing, or by way of plasticity. In the current study, we use theory and reaction-diffusion modeling to show how the first condition may be achieved, and develop detailed multi-scale models to address the second and third.

## Results

As the setting for this study, we considered a network in which ensembles of neurons are active in sequence, for example, place cells in the hippocampus as an animal moves along a linear track (*Figure 1*). We assumed that a single neuron from each of these ensembles projects onto a given post-synaptic cell, which is the focus of our analysis. The projections are ordered such that they converge onto a succession of spines located on a short stretch of dendrite on this cell, in the same spatial and temporal order as the activation of the ensembles. We ignored all other network context. We first treated this system in the abstract, as sequentially ordered activation of segments of a one-dimensional reaction-diffusion system with abstract chemistry. We asked if the reaction system could discriminate sequential from scrambled input. We then mapped the discrimination mechanism to a signaling pathway modeled as mass-action reaction-diffusion chemistry in a similar cylindrical geometry, but ornamented with dendritic spines. We then tested discrimination when we embedded mass-action chemistry into a morphologically and electrically detailed neuronal model receiving sequential input on a series of synapses, through ligand-gated ion channels. Finally, we asked if ion channel modulation by sequence discrimination chemistry in small dendritic zones, could affect neuronal firing.

### Abstract reaction-diffusion systems support sequence recognition

We first analyzed the requirements for chemical reaction-diffusion systems to achieve selectivity for spatially and temporally ordered inputs. In doing these calculations, we utilized 'spherical cow' models of chemistry: highly reduced formulations with just two diffusive state variables A and B, interpreted as molecules undergoing 1-dimensional diffusion (*Figure 2A,B*). The reaction system received input from a third molecule, designated as Ca (*Figure 2A,B*). In all cases, molecule B inhibited A.

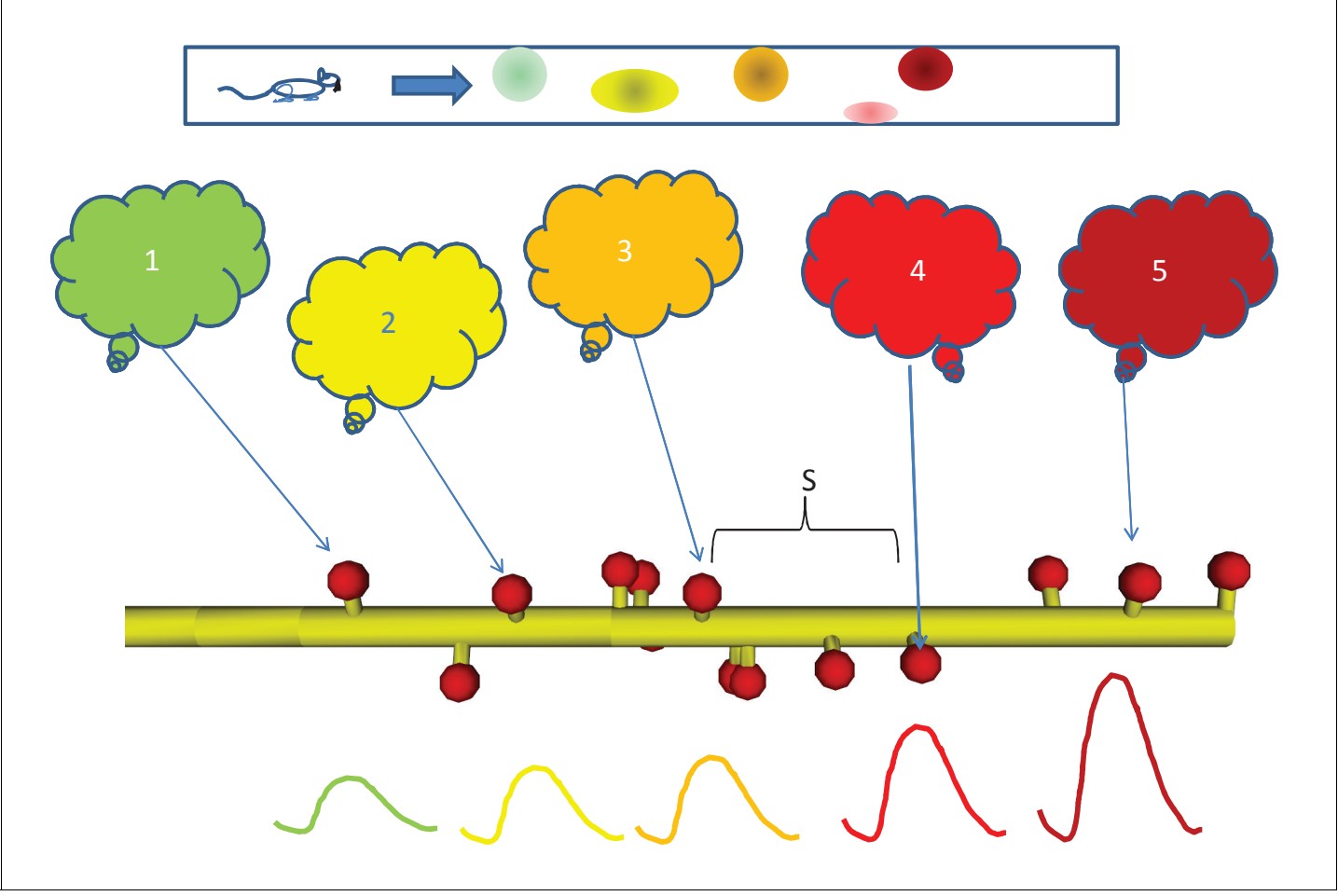

**Figure 1.** Sequential activity, from behavior to dendrite. Top: Linear arena in which a rat moves, with indicated locations of 5 place cells (color coded green, yellow, orange, red, and maroon in order.). The place cells are representatives of 5 neuronal ensembles (colored clouds), each active in one of the five locations on the arena. The neuronal ensembles each send a single axonal projection in spatial order to a small dendritic segment on a postsynaptic neuron, with an average spacing $S$ between connected spines. Note that spines need not be immediately adjacent to each other. Below, buildup of reactant following input activity with the appropriate timing, corresponding to the rat moving at a speed which the dendritic chemistry recognizes.

The abstract models were initially formulated to implement reaction-diffusion wave propagation at a speed that matched the arrival of sequential input. We implemented two models with this property: the FitzHugh-Nagumo form, which is a known stimulus-triggered, oscillating, and propagating wave system (*Fitzhugh, 1961*; *Nagumo et al., 1962*); and a bistable reaction-diffusion system, which again is a known substrate for propagating waves (*Keener and Sneyd, 1998*). To the bistable switch system, we added inhibitory feedback to restore the switch to baseline after a delay. In addition to these models with complex dynamics, we implemented a system with feedback inhibition, and a system with feedforward inhibition. These simpler models were designed to test if inhibitory feedback or feedforward systems, coupled with diffusion, could achieve selectivity in the absence of an amplifying positive feedback process.

Our criterion for sequence recognition was that ordered inputs should elicit high total activity of a signaling molecule A, whereas scrambled inputs should elicit low levels of activity. Total activity here means the sum of activity of A over the duration and spatial extent of the stimulus.

To investigate the temporal responses of these 'reaction' systems, we first delivered impulse (red dot) and step-function (green line) input to non-spatial versions of the models (*Figure 2C*). Each model exhibited a large impulse response. Two of the models (feedforward inhibition and state-

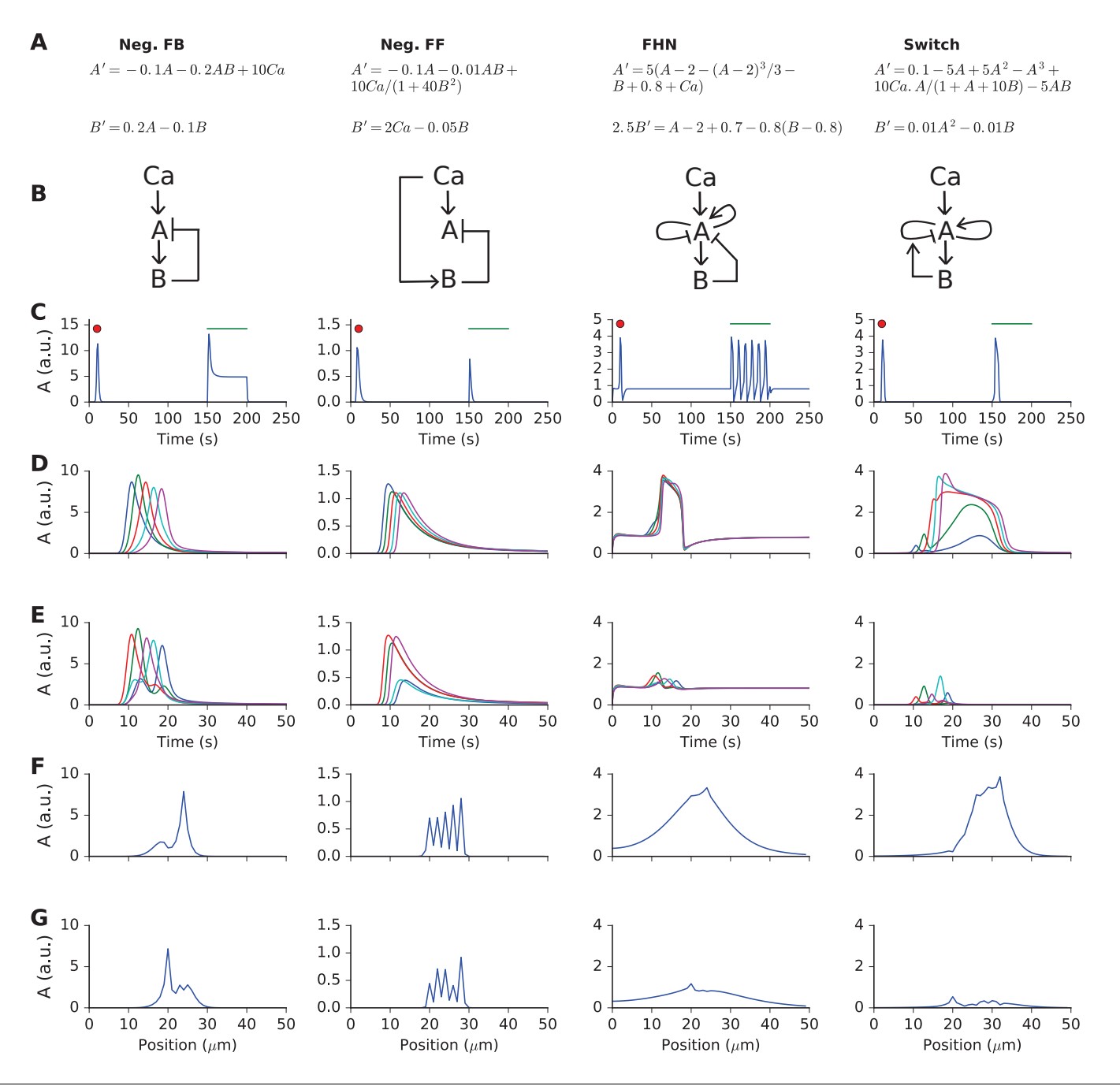

**Figure 2.** Responses and sequence selectivity of four abstract models involving molecules A and B. Columns are the respective models. (**A**) Model equations. (**B**) Model schematics. Arrows indicate excitation, plungers indicate inhibition. In the FHN model, A has a fast self-excitation and a slow self-inhibition. In the Switch model, we indicate a multiplicative inhibition of A by an arrow from B onto the self-inhibition arrow of A. (**C**) Response of point (non-spatial) model to a 1 s wide Gaussian input of $Ca^{2+}$ (red dot) and a steady pulse of $Ca^{2+}$ (green line). Input amplitudes: 1, 10, 0.4, and 1 respectively. (**D-G**) Response of molecule A in one-dimensional reaction-diffusion form of model to sequential input at five locations. (**D**) Time-course of response at five locations to ordered input. Locations are color-coded in the sequential order blue, green, red, cyan, purple. (**E**) Time-course of response at five locations to scrambled input. Note that the FHN and Switch models have much lower responses to scrambled as compared to ordered input. The Neg. FF model has a lower response at two of its locations. (**F, G**) Snapshot of spatial profile of response to ordered (**F**) and scrambled (**G**) input. Snapshot is at time 18.4, 14.2, 16.3 and 18.4 s respectively.

switching) exhibited only a transient response to the sustained input. The negative feedback model, as expected, had a transient strong response followed by a shallow sustained response. The Fitz-Hugh-Nagumo model, again as expected, oscillated.

We then implemented 1-dimensional reaction-diffusion versions of these models (*Figure 2*, methods). We delivered $Ca^{2+}$ stimuli at five equally spaced points on the reaction system. Each $Ca^{2+}$ stimulus followed a Gaussian time-profile:

$$Ca = exp(-((phase - t)^2)/(2\ width^2))\tag{1}$$

Here, *phase* defines the timing of each input, *t* is time, and *width* is the duration of the $Ca^{2+}$ pulse.

The only difference between the sequential and scrambled stimuli was their order ([0,1,2,3,4] and [2,1,4,3,0], respectively (methods)).

The results of these simulations are compared in *Figure 2D–G*. First, we found that the simple feedback inhibition model showed no input sequence-dependent difference in total activity. This can be seen in terms of time-response at the five stimulus points (*Figure 2D,E*), and also in terms of spatial profile of A at the end of the stimulus (*Figure 2F,G*, *Video 1*).

Then, we observed that the feedforward inhibition model showed a small amount of selectivity (*Figure 2D–G*). This arose because the A response was diminished at two of the input points when the input was scrambled (*Video 2*).

The FitzHugh-Nagumo (FHN) model was strongly selective, with a large buildup of response only when the input was sequential (*Figure 2* third column, *Video 3*). We interpret this as arising when the propagating wave from the FHN equation arrived at successive input points just when the input was also present.

The switching model was the most selective (*Figure 2* last column, *Video 4*). This was because it built up to a large, sustained response lasting several seconds, provided the successive inputs were in the same position as the diffusively propagating activity of A.

Thus, this set of simulations showed that several reaction-diffusion like systems were capable of sequence selectivity. Strong selectivity emerged from a supralinear buildup of responses when diffusively propagating activity was aligned with successive inputs, and suppression due to inhibition when the alignment was off.

## Reaction systems select for distinct speeds and length-scales of sequential input

We next asked how the different reaction systems were tuned to different spatial and temporal intervals. We devised a scalar metric, Q, of the degree of sequential ordering of stimuli, as a first step. We plotted the ordinal position of the input against the ordinal value of its arrival time. Thus, a perfect sequence would arrive at positions [0,1,2,3,4] at times [0,1,2,3,4]. We used the quantity

$$Q = mR^2\tag{2}$$

as the measure of how ordered the sequence was. Here, *m* = slope and *R* = regression coefficient of best-fit line. $R^2$ provides a measure of linearity of the sequence, and *m* assigns it a magnitude and

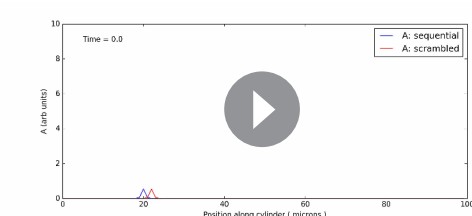

**Video 1.** Time-and-space series of response of negative feedback model to sequential (blue) and scrambled (red) input.

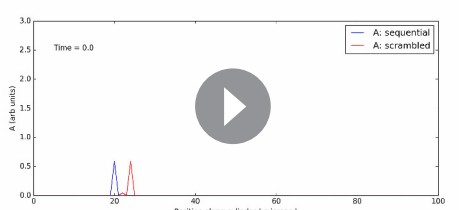

**Video 2.** Time-and-space series of response of negative feedforward model to sequential (blue) and scrambled (red) input.

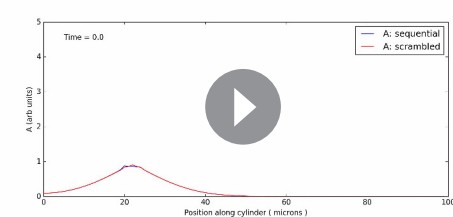

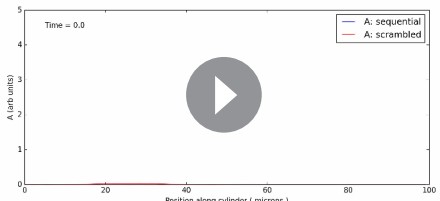

**Video 3.** Time-and-space series of response of FitzHuge-Nagumo model to sequential (blue) and scrambled (red) input.

**Video 4.** Time-and-space series of response of bistable switch model to sequential (blue) and scrambled (red) input.

sign. With this measure, the sequence [0,1,2,3,4] has Q = 1; [4,3,2,1,0] has Q = −1, and [4,0,2,1,3] has Q = −0.001. Examples of regression plots used to generate Q for different sequences are presented in *Figure 3A,B*.

We then compared our reaction readout A with the metric Q, to see how well each of our four reaction systems could discriminate sequence order (*Figure 3C,E,G,I*). Here, we ran simulations with each of the possible permuted input sequences. In each run, we summed the level of A over all five stimulus points, and from the time of the first stimulus till the end of the run, in order to obtain a cumulative estimate of the response. This total A value (*Atot*) was plotted against the Q metric for each of these permuted sequences (*Figure 3C,E,G,I*). If the chemical system was highly selective, we expected that only one or two points in the scatter plot should have a high value of *Atot*, and these should be the points with Q around +1 or −1. Chemical systems that did not exhibit sequence discrimination should have similar values of *Atot* for all sequences.

The results were as expected from the individual runs from *Figure 2*. Specifically, the inhibitory feedback system had poor tuning, there was a small amount of selectivity for the feedforward inhibition case, and strong selectivity for the FitzHugh-Nagumo and switching cases. Each of these outcomes had been observed in *Figure 2* for just the sequential and a single scrambled stimulus, and in *Figure 3* we found that it generalized to the entire set of sequence permutations. Note that the modeled reaction-diffusion system was spatially symmetric and did not distinguish between forward and backward sequences. An analysis of symmetry breaking arising from incorporation of biological detail is out of the scope of the current paper.

We next asked how selective each reaction system became, when the inputs were delivered at different time and space intervals. The time and space intervals were specified as parameters for each simulation run. We devised a metric for reaction network sequence selectivity, based on a comparison of responses to sequential vs. scrambled inputs:

$$Selectivity = (Asequential - mean(Atot))/max(Atot) \qquad (3)$$

Provided the system responds more strongly to sequential input than to any other order, *selectivity* should be in the range from zero (unselective) to one (highly selective). *Asequential* is *Atot* for the sequential stimulus, *mean(Atot)* is the mean of *Atot* over all permutations of the stimulus order, and *max(Atot)* is the maximum value of *Atot* for all permutations of stimulus order.

To compute *Selectivity*, we carried out simulations of the model for all possible non-repeating patterns of input, and obtained *Atot* in each case (*Figure 3*, first column). *Selectivity* was then estimated as per *Equation 3*. We repeated these calculations for each point in the matrix of timing and spacing values, to obtain a grid of *selectivity* values (*Figure 3D,F,H,J*). As already seen from *Figure 2*, the feedback inhibition model had low selectivity in all cases. The feedforward inhibition model showed a diagonal band of weak selectivity such that rapid stimuli in close proximity as well as slower stimuli at greater distances were preferred. The FitzHugh-Nagumo case was much more strongly selective, but also had a similar diagonal band. In both these cases we interpret this as there being an intrinsic propagation speed of the chemical activity wave, and if the inputs were timed and spaced accordingly, the response built up. The switching model was strongly selective, and had a large diffuse zone of strong selectivity centred around (2 s, 4 μm).

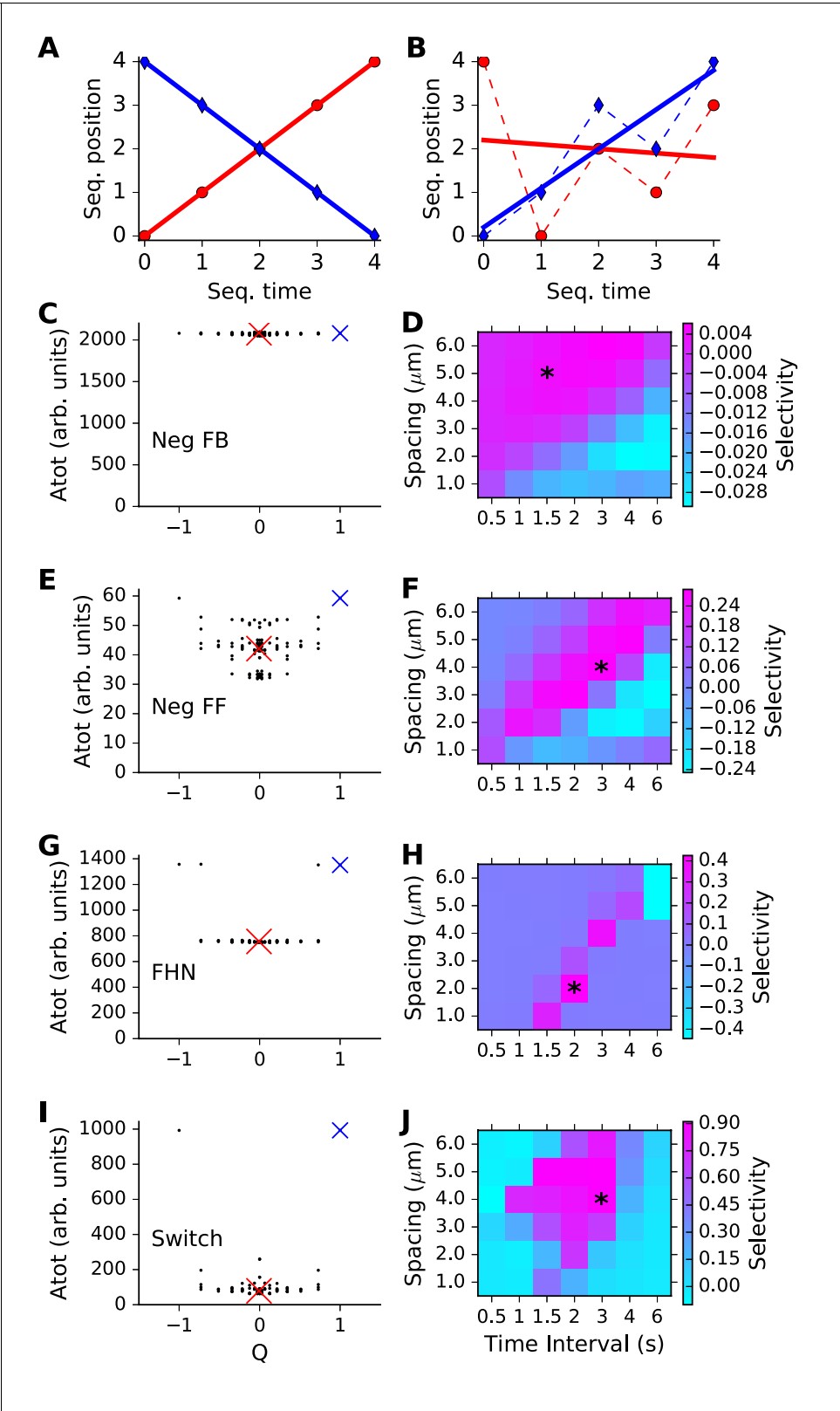

**Figure 3.** Sequence selectivity for different models. (**A, B**) Regression plots for sample sequences. (**A**) Perfect forward sequence [0,1,2,3,4] (red, Q = 1) and reverse sequence [4,3,2,1,0] (blue, Q = −1). (**B**) Scrambled sequence [4,0,2,1,3] (red, Q = −0.001) and [0,1,3,2,4] (blue, Q = 0.729). (**C, E, G, I**) Scatter plots of chemical system selectivity (measured as total activation of molecule A over time and space) against Q. There are 5! = 120 points, each being the outcome of a single simulation with a different sequence. In each plot the blue cross is the perfect forward sequence (whose timeseries is shown in

*Figure 3 continued on next page*

*Figure 3 continued*

*Figure 2D*), and the red cross is the scrambled sequence [2,1,4,3,0], whose timeseries is shown in *Figure 2E*. (D, F, H, J) Matrix of selectivity as a function of total length of stimulus zone and interval time between successive stimuli. In each plot, the asterisk is placed on the matrix entry obtained from the spacing and interval parameters used for the scatter plot to its left. In other words, the selectivity value for that matrix entry is obtained from *Equation 3* using the *Atot* scores from the scatter plot. (C,D) Feedback model. This shows no selectivity. (E, F) Inhibitory feedforward model. Scatter plot in E represents calculations performed at time/distance values of (3 s, 4 mm). This shows low selectivity, as seen by slightly higher *Atot* for Q values of +1 and −1. (G,H) FitzHugh-Nagumo, scatter plot at (2 s, 2 mm). The model is selective in a narrow, diagonal range of time and distance. Its score is somewhat reduced because of the high baseline of *Atot*. (I,J) Switching model, scatter plot at (3 s, 4 mm). This is highly selective over a wide range of time and distance. Only the perfect forward and reverse sequences have high values of *Atot*.

Thus, each of the three sequence-selective reaction systems had a preferred range of stimulus timing and spacing, but the FHN and switch forms had much higher selectivity.

## Sequence speed selectivity scales with reaction rates and diffusion constants

As a further analysis on our abstract models, we asked how sequence speed selectivity scaled with rates and diffusion parameters. This scaling is important because it determines the spatial extent of the sequence selective zones that we propose to exist in the dendritic tree. It is also important as it defines the time-scales of sequential events that may be recognized by such reaction-diffusion mechanisms.

We first tested the most straightforward assumption, that sequence propagation speed was proportional to chemical rate and diffusion constants. To do this, we scaled all rates and the diffusion constants by the same factor, and reduced the stimulus width by the same factor. We ran the same grid of stimulus space and time intervals as above. As expected, scaling all the chemical and diffusion rates also scaled the sequence speed to which the network had the strongest response. As the rates were increased, the best tuning was at shorter time intervals but roughly the same spatial intervals. (*Figure 4A–E*). However, different stimulus strengths were needed to obtain strong tuning in these runs. We therefore also examined dependence of tuning on stimulus strength, and observed that the tuning zone also depended on stimulus strength (*Figure 4F,G*). This is because the selectivity occurs when the stimulus is strong enough that the ordered sequence leads to pathway activation, but not so strong that the stimulus overrides the inhibitory reactions and produces indiscriminate activation. To better understand the range of sequence selectivity, we varied each of the parameters of the model one at a time with respect to the reference model. We carried out this 1-dimensional parameter sensitivity sweep for the FHN and switching models (*Figure 4—figure supplements 1* and *2*). In each case we computed *Asequential* (blue traces), and compared it to *mean (Atot)* averaged over 12 input patterns, 11 of which were scrambled (green traces, methods). We also computed sequence *selectivity* as per *Equation 3* (red traces). We found that selectivity was robust to some parameters, such as diffusion constants, but fragile (range ~20%) with respect to some rate constants and stimulus amplitude. These models were intended to be illustrative and to explore the properties of sequence-selective systems. Hence, we did not optimize the models for robustness.

Thus, by varying rates for reactions and diffusion, we were able to achieve good sequence selectivity over a broad range of time-intervals (0.5–4 s between stimuli) and spatial intervals (2–6 μm between stimuli).

## Mass-action reaction-diffusion system based on MAPK feedback discriminates sequences

We next asked if above design principles for sequence selectivity could be applied to a mass-action reaction-diffusion system. We based our mass-action model on a published, reduced model of MAPK feedback and bistability (*Bhalla, 2011*). The model already exhibited the switch-like turnon behavior of our abstract 'switch' model. The other key aspect of the abstract model was delayed turnoff of activity. There are several known inhibitory feedback turnoff mechanisms for the MAPK pathway (*Lake et al., 2016*). We therefore added a MAPK-activated protein phosphatase to cause delayed turnoff of the kinase. The other changes we made to the published MAPK model were (a) to

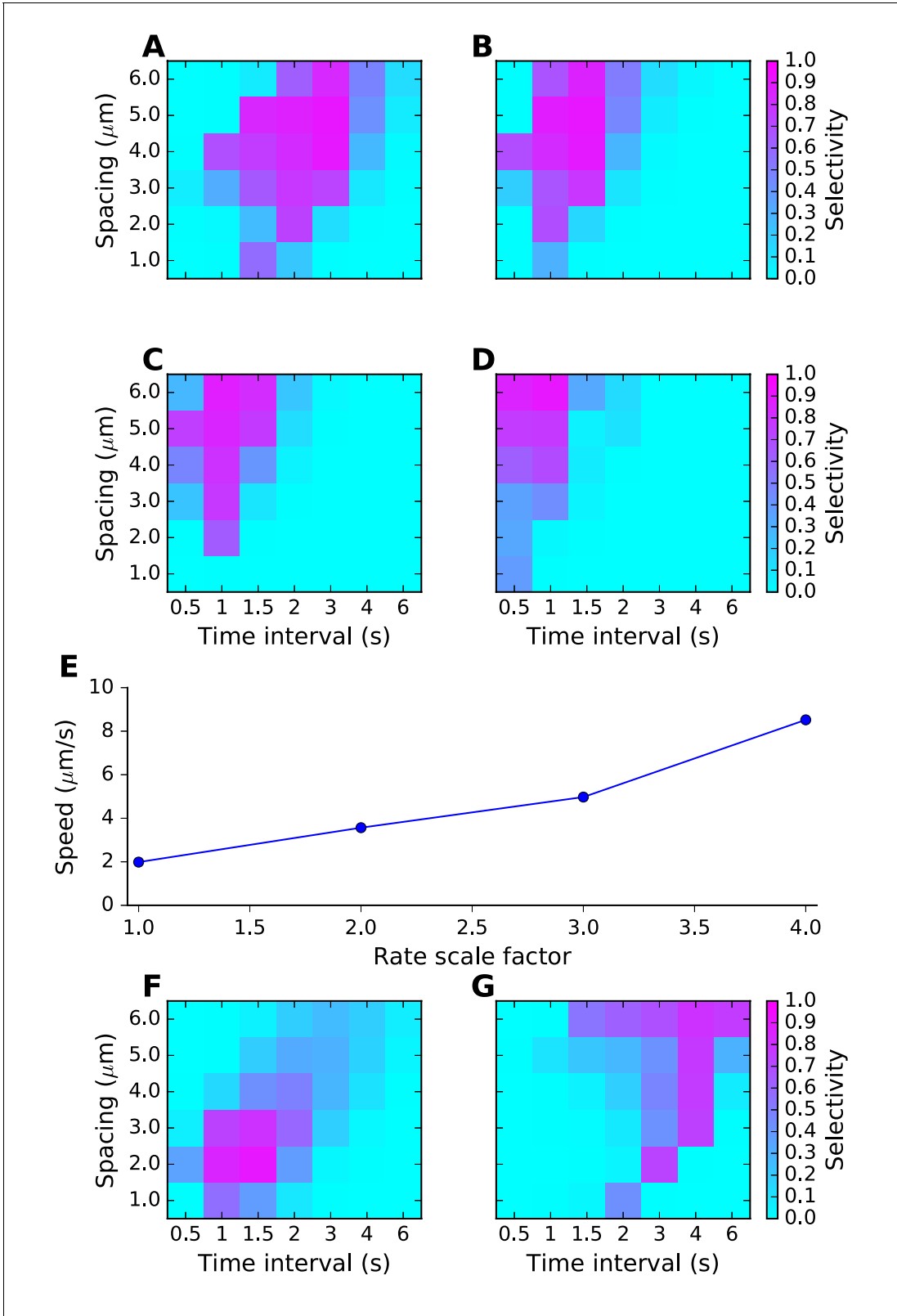

**Figure 4.** Sequence tuning ranges in space and time. (A-D) tuning for rates of 1, 2, 3 and 4 times basal, respectively. (E) Plot of speed of sequence as a function of the rate scale factor. (F, G) Preferred space/time of tuning shifts with stimulus amplitude. (F) 90% basal. (G) 110% basal.

*Figure 4 continued on next page*

*Figure 4 continued*

The following figure supplements are available for figure 4:

**Figure supplement 1.** Parameter sensitivity for FHN (FitzHugh-Nagumo) reaction-diffusion model.

**Figure supplement 2.** Parameter sensitivity for switch (Propagating bistable switch) reaction-diffusion model.

remove the synaptic signaling leading to receptor turnover in the spine and (b) to add diffusible CaM as a $Ca^{2+}$ buffer. With the exception of active PKC, Raf, and the ion channels, all molecules were diffusible. The model schematic is shown in *Figure 5A*, and the full model specification is presented in supplementary material. This model formulation is a semi-quantitative map to the far more detailed and tightly constrained models of this pathway in the literature (*Bhalla and Iyengar, 1999*; *Resat et al., 2003*). $Ca^{2+}$ stimuli were delivered in the PSD and diffused into the dendrite (Methods). Model responses to brief (1 s) and square step function (100 s) $Ca^{2+}$ inputs are shown in *Figure 5B, C*, both measured in the dendrite. As expected, there is a strong switch-like turnon to $Ca^{2+}$ stimuli, followed by delayed inhibition. We ran this model in a cylindrical geometry ornamented with spines at ~1.1 µm intervals (Materials and methods, *Figure 5D*). We provided sequential and scrambled $Ca^{2+}$ input to 5 of the spines separated by ~3 µm each, and observed good sequence selectivity (*Figure 5E,F*).

Thus, we showed that a mass-action reaction-diffusion system exhibited sequence selectivity when it had switch-like turnon with delayed turnoff by negative feedback. This configuration had been predicted by the abstract models.

## Sequence selectivity works in a biologically detailed multiscale neuronal model with noisy input

We then asked whether biochemical sequence recognition would 'work' in the more complex context of active neurons receiving background activity in a network. We brought in biological detail at the following levels: (a) Model neuron morphology based on anatomical reconstructions (>3000 segments), including >5700 spines spaced at ~1 µm. (b) Voltage-gated ion channels distributed throughout the cell, based on published models. (c) Background glutamatergic Poisson synaptic input at 0.1 Hz on all spines. (d) Background GABAergic synaptic input with an 8 Hz theta modulation on proximal dendritic compartments. Due to the background synaptic input, the model exhibited theta-modulated sub-threshold oscillations, with spiking activity at ~1 Hz (*Figure 6B*). (e) Stimuli in the form of spike trains arriving on AMPA and NMDA receptors on subsets of spines. (f) $Ca^{2+}$ dynamics following synaptic ion flux, including diffusion between spine and dendrite, and buffering by calmodulin, in all spines and dendrites (*Video 5*). (g) The reduced MAPK model described above, was distributed throughout the dendritic tree in ~6000 diffusive compartments (*Figure 6A*). (h) Chemical calculations in the spines were carried out using a stochastic method (Gillespie Stochastic Systems Algorithm, methods) for the runs used to calculate selectivity.

We first ran a direct comparison of a sequential input train [0,1,2,3,4] compared with a scrambled train with sequence [4,0,3,1,2]. We picked time and space intervals of (3 s, 4 µm) based on preliminary calculations for tuning. We delivered the input sequence on a basal dendritic branch (*Figure 6A*). We selected a set of 5 spines, spaced at ~3 µm. Each spine was stimulated with a Poisson spike burst lasting 2 s at 20 Hz on the AMPA and NMDA receptors. With these parameters, the $[Ca^{2+}]$ reached ~40 µM in the PSD, ~35 µM in the spine head, and ~1 µM in the region of dendrite immediately below the stimulated spine (e.g., *Figure 6C*). Similar $Ca^{2+}$ levels were obtained when stimuli were delivered in other regions of the dendrite. The exception was the primary apical dendrite, in which dendritic $Ca^{2+}$ only reached ~0.5 µM. This is an expected outcome of diluting out the $Ca^{2+}$ arriving from the spine, since the diameter of the primary dendrite was ~2 µm as compared to ~1.0 µm in the other stimulus regions. With these parameters, we observed strong selectivity in MAPK activity for sequential stimuli as compared to scrambled in the basal dendrite (*Figure 6D*, solid lines for sequential vs. dashed for scrambled).

We then asked how this multiscale model responded to a range of time and space intervals in each of the stimulated zones. Because the calculations were expensive, we used a modified version

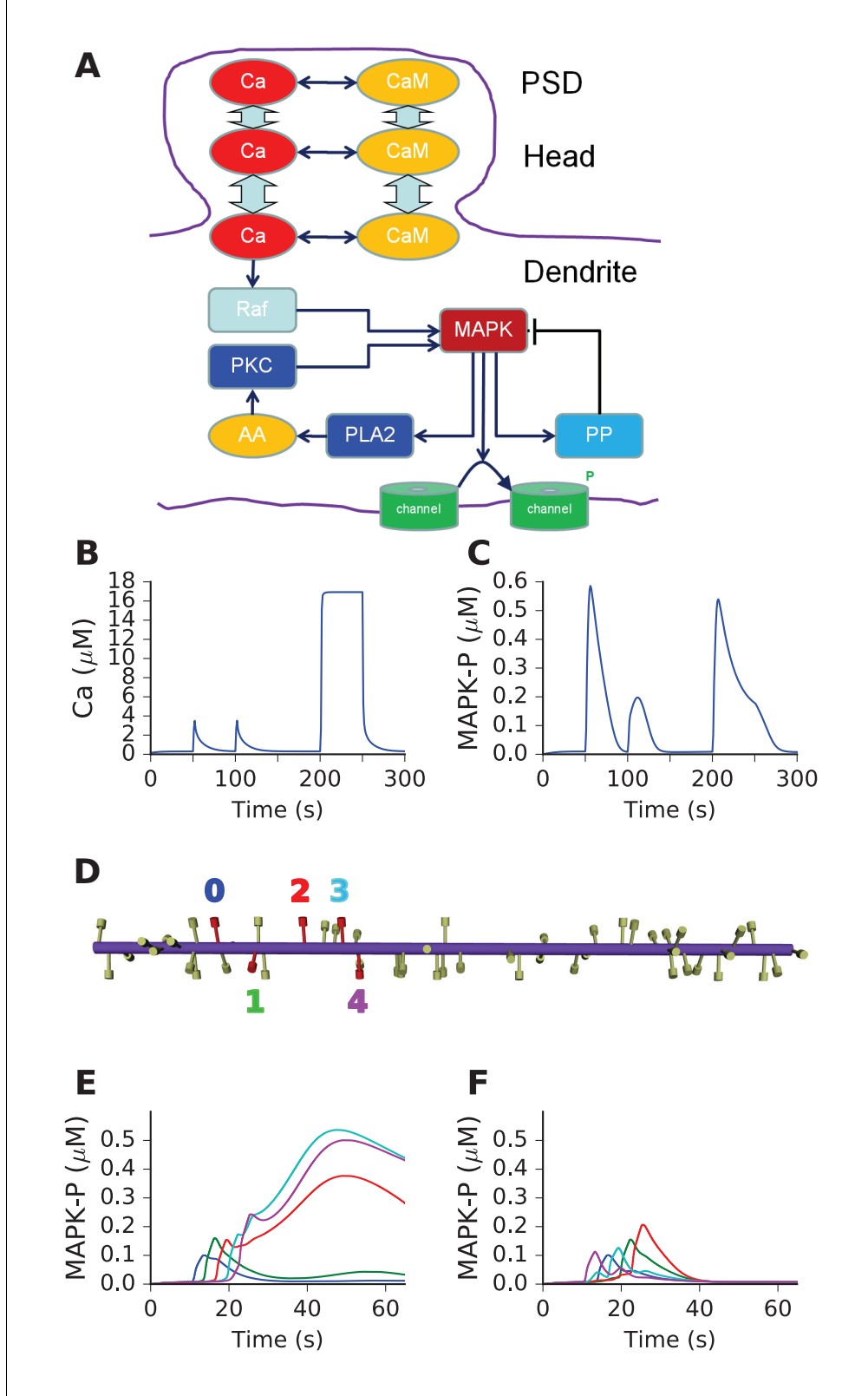

**Figure 5.** Mass action model of a MAPK switch exhibits sequence selectivity. (**A**) Schematic of model chemistry. Black arrows indicate binding or activation reactions, plunger indicates inhibition, and bent arrow indicates enzyme activity. Broad cyan arrows indicate diffusion of $Ca^{2+}$ and CaM between PSD, spine head, and dendrite. CaM buffers the incoming $Ca^{2+}$ from the PSD. (**B**) $Ca^{2+}$ responses seen at the dendrite, to input delivered at PSD. PSD stimulus was two pulses of 1 s, 160 mM $Ca^{2+}$, separated by 50 s. After a delay of 100 s, a step of 160 mM $Ca^{2+}$ was delivered for 50 s. (**C**)
*Figure 5 continued on next page*

*Figure 5 continued*

MAPK-P response to the Ca$^{2+}$ stimulus. Note strong inhibition of the response to the second pulse, and rapid decline of response to the 50 step stimulus. (**D**) Geometry of 1-D reaction-diffusion model, with 49 spines. Dendrite diameter was 1 μm and length was 60 μm. The five stimulated spines are in red, each identified by a color coded number. (**E**) Response of system to sequential input [0,1,2,3,4] on indicated spines spaced ~3 μm apart, at intervals of 3 s. Each input was delivered to the PSD of the stimulated spine, at 160 μM, for 2.9 s. MAPK-P was recorded from the dendrite subdivision attached to each of the stimulated spines. Plot colors indicate stimulated spine from D. (**F**) Response to scrambled input, in the order [4,0,3,1,2]. The response to scrambled input had a smaller amplitude and lasted for a shorter time. Plot colors are as per spine numbers in D.

of the selectivity metric (*Equation 3*) in which we computed outcomes of only 12 stimulus patterns rather than the exhaustive 120 possible permutations (Materials and methods). In each case, we used MAPK-P in place of the molecule 'A' for calculating selectivity (*Equation 3*). For efficiency, we delivered the stimulus patterns simultaneously in the 4 zones of the cell (*Figure 6A*). Similar tuning was observed in test simulations where stimuli were delivered only in a single zone.

We observed strong sequence selectivity in the basal dendrite zone, for a restricted range of space and time intervals, as expected from the earlier calculations (*Figure 6E–G*). We repeated the calculations using different random seeds to generate different background and stimulus spiking input to the model neuron. We found that the responses of individual runs, and hence selectivity, were somewhat noisy (*Figure 6E–G*).

We next investigated whether the response selectivity was manifested in different regions of the cell. We found that there were indeed strongly sequence selective responses in oblique dendrites and distal apical dendritic regions, but these required stronger stimuli (*Figure 6H,I*; *Videos 6* and *7*). In these runs, we had to increase the diameters of spine heads and spine shafts by 40% and 20% respectively to obtain sufficient dendritic Ca$^{2+}$ influx, and hence selectivity. The reference spine models had a shaft of 1 μm length x 0.2 μm diameter, and a head of length and diameter 0.5 μm. We did not observe sequence recognition in the primary apical dendrite (*Figure 6J*). Even the strongest stimuli applied in this location elicited only small Ca$^{2+}$ elevations, with weak downstream MAPK responses, and no pattern selectivity.

We performed a limited parameter sensitivity analysis for the detailed cell model, focusing on biophysical and biochemical parameters involved in triggering or responding to Ca$^{2+}$. We chose these parameters since stimulus range had been one of the most sensitive parameters for the abstract models. We individually varied stimulus strength, L-type Ca$^{2+}$ channel density, Ca$^{2+}$ diffusion, and the activation of Raf by Ca$^{2+}$. Each of these runs was performed with distinct random seeds and stochasticity in spike-trains and in chemical kinetics in spines. Even though we had not fine-tuned the model for robustness, we found that the model retained selectivity (at least 50% of best selectivity) over a factor of ~2 or more for all parameters except the Raf affinity for Ca$^{2+}$ (*Figure 6— figure supplement 1*). The latter parameter could be varied by ±15% about its reference. While this list is not exhaustive, it does suggest that the sequence selectivity of the detailed model is fairly robust, and somewhat more so than the abstract models.

Thus, we showed that intracellular signaling pathways exhibited selectivity for ordered synaptic input, even when considerable morphological and electrical detail and noisy input were incorporated. Selectivity became noisier with this additional detail, and different zones of the cell required different synaptic efficacies to achieve selectivity. With the current parameters the selectivity was on the time-scale of seconds, with successive inputs spaced apart by a few microns.

## Sequence-signaling triggered channel modulation may influence cellular firing

There are multiple possible outcomes of local sequential activity-triggered chemical signaling. These include synaptic plasticity, dendrite remodeling, and electrical changes through channel modulation. For the purposes of the current study, we focused on the third, namely channel modulation leading to changes in electrical activity, since this defines how the neuron may report the presence of a sequence to other cells in the neuronal network. Similar brief, strong synaptic input on small sections of distal dendrites has been reported to cause strong changes in activity of striatal spiny neurons by triggering a transition to an 'up' state (*Plotkin et al., 2011*).

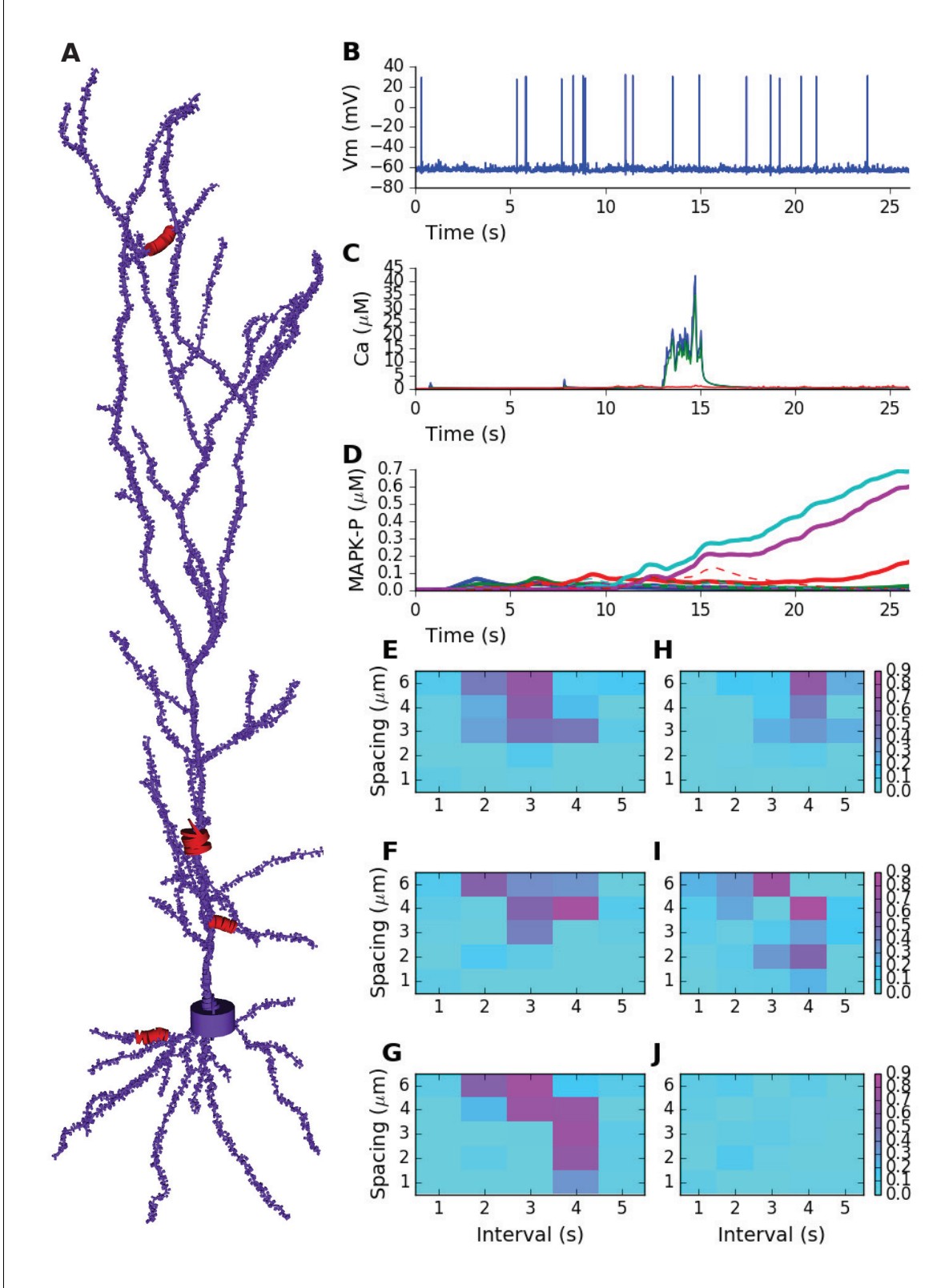

**Figure 6.** Sequence selectivity in detailed electrical+chemical signaling model. (A) Morphology of model. Dendrite diameters are scaled up by 2, and spine diameters and lengths by four for visualization. Stimulus was given in four zones on the cell, indicated in red and by diameters scaled up by 10x. (B) Example somatic intracellular potential and spike train of model neuron. (C) Ca$^{2+}$ responses to Poisson synaptic volley at mean rate 20 Hz, measured in PSD (blue), spine head (green) and dendrite (red). (D) Sequence selectivity in the distal apical zone. Heavy solid lines are P-MAPK-P levels under the

*Figure 6 continued on next page*

*Figure 6 continued*

stimulated spines, for sequential input. Strong buildup occurs for three of the spines. Dashed lines are corresponding MAPK-P levels for scrambled input. Only the sequential input responses lead to build up. (E-G) Matrix of sequence selectivity in basal dendrite zone, for different input spacing in time and space. Three different runs are shown, with the same morphology but different random number seeds for the background and stimulus synaptic input. (H) Selectivity in proximal oblique dendrite zone, using 40% larger spine dimensions. (I) Selectivity in distal apical dendrite using 20% larger spines. (J) Selectivity matrix in proximal primary apical dendrite shows no sequence selectivity even with 40% larger spines.

The following figure supplement is available for figure 6:

**Figure supplement 1.** In each plot, the blue trace is the summed concentration of MAPK-P for the sequential stimulus, and the green plot is the mean of summed MAPK-P over 12 permutations of the stimulus.

We utilized the same morphologically detailed pyramidal neuron model as above, and asked if local channel modulations over the length scale of about 20 μm could alter somatic spiking. We assumed that MAPK activity, or other signaling events downstream of sequence selective chemistry, could perform channel modulation through phosphorylation or triggering of channel insertion. For example, MAPK phosphorylates and modulates KA (*Yuan et al., 2002*) Kir6.2 (*Lin and Chai, 2008*), and Na(v) 1.7channels (*Stamboulian et al., 2010*). MAPK is also implicated in control of trafficking of AMPA receptors (*Keifer et al., 2007*). We did not simulate these modulation events explicitly since the computations were very lengthy and we wished to perform many repeats to estimate the distribution of the firing rates following channel modulation. Instead we ran just the electrical cell model for an initial settling period of 1 s, and then used the simulation script to modify the selected channel conductance to represent its modulation. Following this we ran the model for another 3 s in each case. We applied these channel modulations to the same four regions of the cell which we had tested for sequence discrimination. We examined modulation of KA, Na, nonselective leak, NMDAR and AMPAR channels.

We estimated the statistics of firing following these modulatory changes through repeated simulations of the control condition (100 repeats) and each modulation condition (40 repeats), each with different random number seeds. We found that modulation of the Na channel, and increase in nonselective leak conductance in the apical and primary dendrites could increase firing rates by large factors (*Figure 7A,B*).

In initial runs, modulation of NMDA and AMPA receptors had no effect. This was because the random background synaptic input to these receptors was low, at 0.1 Hz. Since the receptors were activated at this low rate, their effect on cell firing was minimal. However, in these calculations, we had omitted the synaptic input that triggered the chemical cascades. Based on the time-course of build-up of MAPK-P (*Figure 5E*, *Figure 6D*), the MAPK-P activity was already high by the time of the last synaptic burst in the sequence of inputs. We therefore repeated the calculations, and additionally incorporated a single burst of synaptic input to the NMDA and AMPA receptors for the same duration and frequency as in the full multiscale cell model (*Figure 6*).

With this change, we found that a 10x increase in AMPAR conductance in the 20 μm zone in the apical dendrite led to a doubling in cell firing (*Figure 7C*). A similar manipulation in the primary dendrite led to a shallower increase, about 1.5x.

Thus, chemical signaling, even in rather small regions of the apical dendrite, may lead to a rapid change in cell firing by local channel modulation. However, large channel modulations were needed to elicit sufficiently large (e.g., 50%) changes in in firing rate that could be detected over the noisy background.

## Discussion

We postulate that recursive sequence recognition is a fundamental computational operation in the brain, and that this operation may be implemented by neurons able to recognize spatially and temporally sequential input on behavioral time-scales (seconds). In this study, we have examined three key parts of such a hypothesis. (1) At the chemical level, we have examined a set of abstract rate equations coupled by linear diffusion, and identified key motifs that support sequence recognition. (2) At the dendrite level, we have shown that these abstract principles of reaction-diffusion mediated

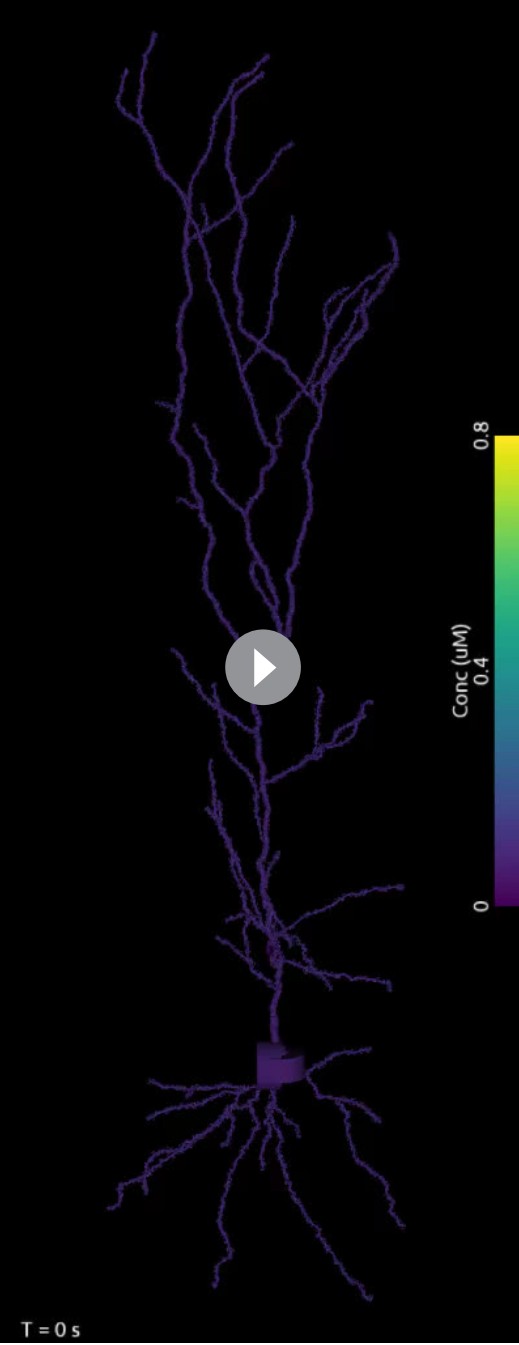

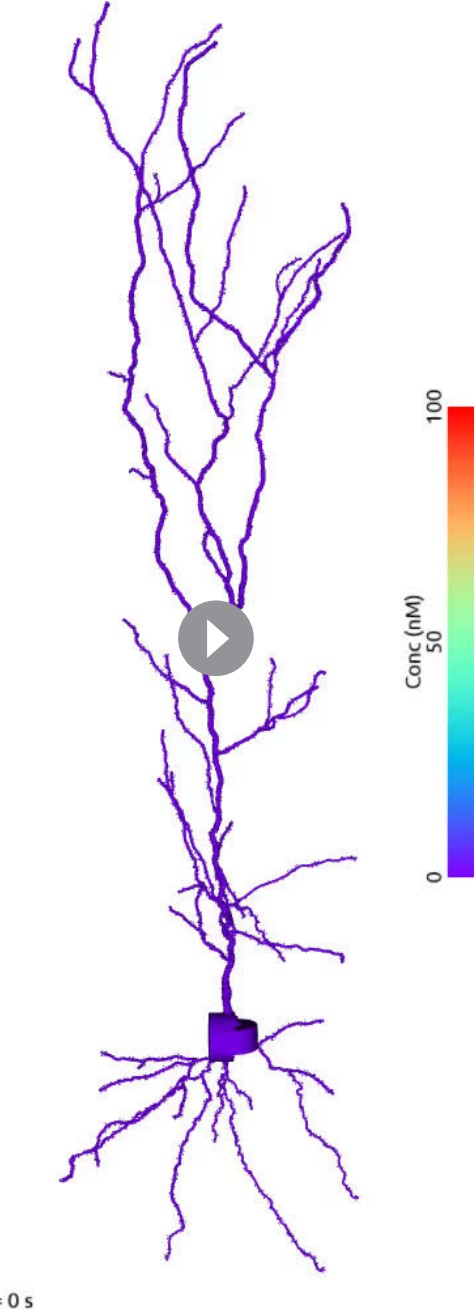

**Video 5.** Calcium influx into spines and dendrites due to random background and sequential stimulation in detailed single-neuron model.

**Video 6.** MAPK-P activity in apical dendrite zone following sequential synaptic input. Note high and sustained buildup of MAPK-P activity in a small zone of ~25 microns.

sequence recognition carry over to much more detailed and biologically motivated neuronal models. These models are multiscale composite models, and explore the length-scales and physiological detail relevant to the current hypothesis. (3) At the cell level, we have shown that local, 20 μm channel modulation can elicit changes in somatic spiking. 20 μm is within the range of our predicted sequence-activated signaling that could lead to channel modulation by phosphorylation or insertion.

We envision a network configuration where at the input level, there are ensembles of neurons that are active in a defined sequence in some sensory, motor or other context. We focus our analysis on a specific postsynaptic neuron, which receives a projection from each of the preceding ensembles, such that each synapse is located within a few μm of the next.

## Firing change, synaptic plasticity, and morphological change may result from biochemical sequence recognition

There are several synaptically activated pathways that may fit the profile of state switching with feedback inhibition. Based on our abstract models (*Figure 2*), these are candidates for implementing sequence selectivity. These include four major kinases (PKA, PKC, MAPK, CaMKII) and a variety of second-messenger and metabotropic pathways (*Bhalla and Iyengar, 1999*; *Kim et al., 2011*; *Lisman and Zhabotinsky, 2001*). Thus, though we illustrate our findings using the MAPK pathway, we suggest that other mechanisms may also apply. Biochemical events have multiple outcomes on cell physiology, and even with the one example of MAPK pathway output we suggest that these may (at least) include firing rate changes, plasticity, and morphological change.

On the rapid time-scale of seconds, MAPK modulates multiple ion channels in the cell. These include KA (*Yuan et al., 2002*), Kir6.2 (*Lin and Chai, 2008*), Na(v)1.7 channels (*Stamboulian et al., 2010*), and MAPK also influences trafficking of AMPA receptors (*Keifer et al., 2007*). As analyzed in *Figure 7*, modulation of some of these target channels can lead to immediate changes in cell firing. While the amount of modulation required for an effect is quite large in some cases, we stress that these were proof-of-principle calculations. A more exhaustive calculation would require better cellular physiology including more channel types, full multiscale calculations of the biochemistry leading up to channel modulation, and multiple simultaneous targets. Similar small length-scale events have been shown to affect cellular firing in striatal neurons (*Plotkin et al., 2011*), suggesting that this outcome of sequence selection may also be plausible. It has also been shown that local signaling can lead to changes in dendritic excitability, again modulating cell-wide firing (*Narayanan and Johnston, 2010*).

On a slightly longer time-scale, MAPK is upstream of numerous plasticity events (*Smolen et al., 2006*). These include phosphorylation and insertion of receptors (*Keifer et al., 2007*), modulation of dendritic protein synthesis (*Tsokas et al., 2007*), and transcriptional control (*Rosenblum et al., 2002*). The computational outcome of synaptic plasticity has been extensively analyzed, but to our

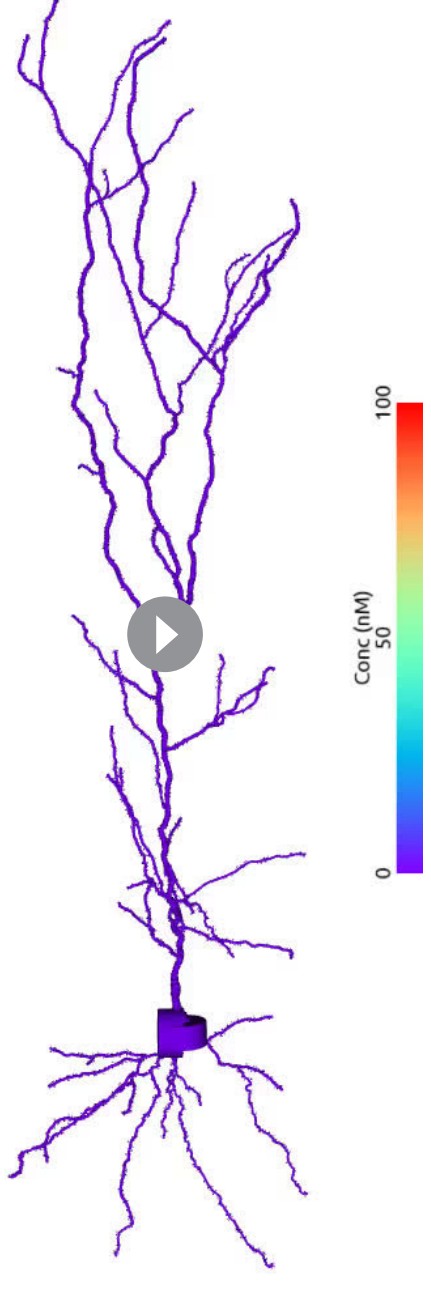

T = 0 s

**Video 7.** MAPK-P activity in apical dendrite zone following scrambled synaptic input. Note that the activity is low and short-lasting.

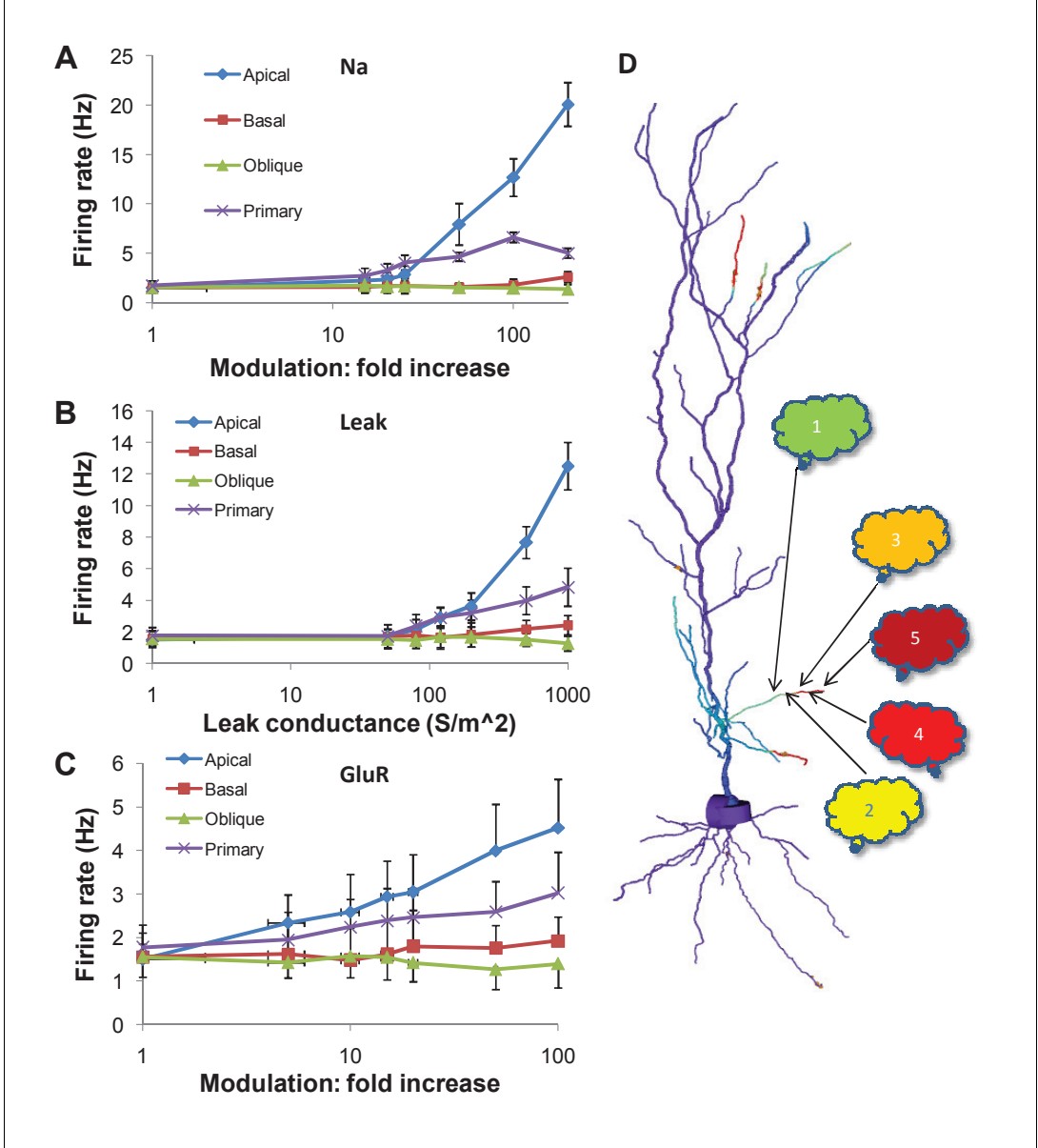

**Figure 7.** Firing rate changes in response to channel modulation in small zones on the dendritic tree. Error bars are standard deviation. (**A**) Modulation of Na current. (**B**) Adding nonselective Leak current. (**C**) Modulation of GluR in the presence of the same synaptic input as was used as a stimulus for the sequential input. (**D**) Schematic of convergence of inputs from different ensembles onto one dendritic zone. The numbered clouds represent input ensembles from which a single neuron projects to the indicated location on the dendrite.

knowledge there are no learning rules that factor in sequential activity. It is intriguing to note that our analysis of dendrite-based local chemical activation has parallels with the phenomenon of synaptic tagging (*Frey and Morris, 1997*). In both cases, synaptic input can amplify events in nearby, weakly-stimulated synapses. We speculate that sequence recognition and its conversion to plasticity events may share mechanisms with synaptic tagging.

Finally, MAPK activation is known to play a role in learning-associated morphological change in neurons (*Tyler et al., 2002*). This raises the possibility that its activation through sequence-selectivity rules may lead to addition of new synapses, or reconfiguration of dendrite geometry and spine placement. Such changes would affect the same reaction-diffusion events that underlie sequence selectivity, leading to interesting multiscale feedback dynamics.

## Sequence recognition confers substantial computational capabilities on single cells

Our model of sequence recognition in the neuron implements a potentially powerful form of cellular computation, leading to multiple possible outcomes as discussed above. It is interesting to estimate how much computation this represents. For simplicity, we consider computation in terms of immediate firing rate changes (discussed above), though plasticity and morphological change are also computational outcomes.

If we take a 5-stage sequence, a perfect sequence recognizer would distinguish 1 from 5! possible sequences, or 1 in 120. Longer sequences would have exponentially steeper discrimination ratios, but this places increasingly stringent constraints on network connectivity due to the requirement of having the inputs close to each other. Based on our simulations, biochemical reactions in the presence of noisy input are not as clean in their sequence selectivity as the abstract models (*Figure 6* vs. *Figure 3*) but nevertheless do achieve good discrimination. In addition to being able to discriminate an ordered sequence from among many others, it is desirable that there be a large difference in chemical signal amplitude between ordered and scrambled sequences. *Figures 2*, *3* and *5* suggest that ratios in the range of 1:5 may be achievable, and 1:2 should be common.

Next, we estimate how many such sequence recognition blocks there might be in a neuron. For this analysis, we draw on the results of the biochemical calculations for an estimate of a sequence recognition zone, of ~10 μm (*Figure 6*), and the result that sequence discrimination should happen over most of the dendritic tree except for the trunk of the primary apical dendrite (*Figure 6J*). Say the set of spines occupies 10 μm. The most conservative calculation requires that these blocks do not overlap, so in a pyramidal neuron with about 10000 μm of dendritic length we have ~1000 blocks. If we permit blocks to slide by one input at a time, there are five times as many sequence recognition blocks, i.e., ~5000. If the sequence lasts 10 s, a single block performs 0.1 sequence recognitions per second. Thus, the neuron may perform 500 sequence discriminations each second. Calcium-induced Calcium release (CICR) based sequence logic (discussed below) is much faster, and may be able to increase this by a factor of 10. It is out of scope of the current study to estimate how many of the 500 possible sequential inputs would be active and in order at any given instant, but it is clear that the outcome of such sequence discrimination across the cell would be much more selective than a simple sum of the synaptic inputs that underlie the sequences. Leaving aside the plasticity effects, the readout of these sequence discriminations could be an ongoing modulation of cellular firing.

A still more speculative calculation suggests that single-neuron sequence recognition may provide a way to handle the combinatorial explosion of possible input sequences. Assume that sequential inputs converge to 10 μm stretches of dendrite, having spine spacing $\sigma = 0.5$ μm. Assume that the sequence-recognition biochemistry tolerates a spatial slop of ±1.5 μm, or ±3 synapses, for each input. Then there are ~7 possible synaptic inputs for each stage of the sequence, and the overall block of 5 inputs may receive $7^5$, or ~17000 sequences of length 5. Using our calculation above, each neuron has ~5000 blocks, and so can recognize ~$8.5 \times 10^7$ sequences. Thus, with many assumptions about connectivity and sparsity, this mechanism of single-neuron sequence recognition suggests that single neurons may recognize very large numbers of input patterns through the combinatorics of synaptic convergence.

The key distinction between these two calculations is that the first indicates the computational capacity of a neuron, whereas the second considers the diversity of inputs it can act upon. Overall, our study suggests that sequence computation in the time domain, coupled with the combinatorics of spatially organized input along dendrites, result in extremely parallel and efficient computation at the single-neuron level.

## Single-cell sequence recognition may act on a continuum of timescales

Neural activity sequences occur at a range of timescales. A striking example of this is the mapping of behaviorally-driven place-cell sequences (seconds) to rapid, 100 ms time-scale replay sequences (*Wilson and McNaughton, 1994*). The distinctive aspect of our current analysis is that it works for slow behavioral time-scales of seconds, which is typically challenging for electrical network computations (but see [*Barak and Tsodyks, 2006*; *Goudar and Buonomano, 2015*]). Our analysis suggests that chemical mechanisms can also support faster sequence recognition on the same length-scales

(*Figure 4*). One way this might be implemented in the cell could be a faster kinase cascade than MAPK, such as PKA or PKC (*Bhalla, 2002*). Another might be local CICR, which has already been proposed to support propagating wave activity in dendrites (*Hagenston et al., 2008*; *Kapur et al., 2001*; *Larkum et al., 2003*; *Lee et al., 2016*; *Plotkin et al., 2013*; *Ross, 2012*) These have faster dynamics, of the order of 100 μm/s. Further, dendritic $Ca^{2+}$ imaging suggests that the length-scale of dendritic CICR is similar (~10–20 μm) to that envisaged for sequence recognition in our study (*Hagenston et al., 2008*; *Kapur et al., 2001*; *Larkum et al., 2003*). Simulation studies have previously suggested that CICR may be a mechanism for integration of inputs and resulting in graded, persistent activity (*Loewenstein and Sompolinsky, 2003*). This study has some parallels with our analysis, in particular the presence of a propagating wavefront of chemical ($Ca^{2+}$) activation. However, in the earlier CICR model the wavefront propagation was modulated by the summed synaptic input to the neuron rather than local and specific sequential synaptic input, which is the basis of our study.

Individual chemical implementations of sequence recognition in our study operate over a range of about 2–5 μm and 1.5–4 s (*Figures 4* and *6*). While this is fairly robust as seen from a parameter sensitivity viewpoint, it falls far short of time-invariance. Rate modulation or stimulus amplitude may shift this to some extent (*Figure 4*) but a given cell, in a given state of modulation or activity, will recognize sequences only within a relatively small range. One possible way to extend the range is to have multiple chemical pathways each carry out sequence discrimination in parallel. It would be interesting to see how the chemistry of multiple sequence recognizers may interact. Alternatively, time-invariance may not be a feature of single neurons, but may arise in a network where different neurons recognize different time-ranges. Different cell-types with different chemical mechanisms for sequence-recognition would be likely to operate in substantially different time regimes. It is interesting to speculate that there may be useful computational implications of having different cells 'tuned' to different sequence speeds.

Still faster sequence recognition (~40 ms) has been demonstrated in the electrical domain, where forward sequences can be discriminated from backward (*Branco et al., 2010*). This electrical mechanism has been shown to have ~40% discrimination between forward and backward patterns, rather than the strong selectivity among permutations of patterns shown by our chemical system. Thus this form of electrical sequence recognition is likely to operate in different network contexts than our proposed chemical recognition mechanism.

Together, we suggest that chemical and CICR-based sequence recognition may span the range of timescales from 200 ms to 10 s. It is interesting to speculate that these may coexist, thus permitting the same segment of dendrite to recognize slow behaviorally-driven sequences and also much faster replays of the same sequence.

## Sequence recognition is testable

Our hypothesis of single-cell sequence recognition is readily testable using current technology. Its chemical basis may be examined in brain-slice using local agonist application or sequential glutamate uncaging followed by microscopic readouts of activity reporters for candidate pathways. Such experiments have already been done to examine CICR triggered by synaptic inputs converging to adjacent synapses (*Hagenston et al., 2008*; *Plotkin et al., 2013*). These show that there is indeed a buildup of $Ca^{2+}$ along small, 20 μm stretches of dendrite, under suitable stimulus conditions. If local biochemical signaling leads to changes in cellular firing-rate (e.g., *Figure 7*, [*Plotkin et al., 2011*]), then sequential uncaging along with patch recordings should report these. Further, pharmacological experiments in the slice would readily be able to tease apart possible mechanisms.

Another implication of the proposed reaction-diffusion mechanism for sequence recognition is that it suggests mechanisms for coupling genetic polymorphisms and mutations to a specific aspect of neuronal computation. For example, small changes in reaction rates may shift the preferred timescale of recognized sequences (*Figure 4*), but large changes may eliminate tuning altogether. It would be interesting to see if there is a correlation between specific dendritic signaling genes and psychophysical measures of sequence computation.

In summary, we propose a novel computational function of single neurons, to recognize slow sequences on behavioral timescales, and to exploit combinatorics of input projections to carry out such recognition in a massively parallel manner. We have used simulations of abstract chemistry and detailed multiscale neuronal physiology to understand implications and constraints for such

computation to occur. We suggest that this multiscale cellular signaling process may underlie computationally powerful sequence recognition mediated by single neurons.

## Materials and methods

All modeling was carried out using MOOSE, the Multiscale Object-Oriented Simulation Environment (*Ray and Bhalla, 2008*). MOOSE is freely available, open source, and licensed under the GNU Public License version 3. It can be downloaded from moose.ncbs.res.in and GitHub https://github.com/BhallaLab/moose (*Bhalla et al., 2016*). MOOSE utilizes the GNU Scientific Library (GSL) Runge-Kutta-Fehlberg fifth order method for chemical computations, and a custom-written branching 1-D diffusion solver using the backward Euler (implicit) method. Stochastic chemical calculations were carried out using a custom-written optimized version of the Gillespie Stochastic Systems Algorithm. Pseudo-random numbers were generated by the Mersenne twister (*Matsumoto and Nishimura, 1998*). Electrical computations utilized a custom version of the branched nerve equation solution methods described by Hines (*Hines, 1984*). This has been validated by comparison with other simulators (*Gleeson et al., 2010*). Interfaces between chemical and electrical signaling components of the model utilized adaptor classes in MOOSE, which average over spatial and temporal discretization differences between the two methods. For example, electrical calculations (yielding $Ca^{2+}$ values) utilize a much smaller timestep (~50 µs; 5 µm) but typically employ a larger spatial step than chemical calculations (~1 ms, 1 µm). The adaptors synchronized the chemical and electrical models every millisecond using first-order corrections to each model system. Electrical model morphologies were either geometrical (cylindrical) or derived from published cell reconstructions available on Neuro-Morpho.org (*Ascoli et al., 2007*; *Dougherty et al., 2012*). Spines were positioned along dendrites using Poisson statistics with a specified mean spacing between spines (1 µm in the full cell model). Spines were modeled as a cylindrical head (0.5 µm length and diameter) on a cylindrical shaft (1 µm length, 0.2 µm diameter). In some simulations the spine dimensions were scaled up by 20% or 40%. Channel kinetics were derived from *Traub et al. (1991)*. Analysis and plotting was done using Python, NumPy, and MatPlotLib. Simulations were carried out on a variety of Linux workstations, and large calculations were carried out on Linux clusters. Figure generation code for *Figures 2*, *5* and *6* is available as supplementary material.

Abstract reaction-diffusion calculations (*Figures 2–4*) were carried out in a 1-dimensional geometry with spatial discretization of 1 µm. In order to represent reactions occurring in a spatially restricted region in or under dendritic spines, the reaction systems were active only in five equally spaced 1 µm patches, but diffusion took place throughout the 100 µm length of the model.

Selectivity calculations were performed according to *Equation 3*. In this equation, *Atot* is computed for each permutation of the sequence. This is expensive as there are 120 permutations for a sequence of length 5. Therefore exhaustive permutations were only performed for *Figures 3* and *4*. In order to generate sequence selectivity matrices for *Figure 6* and for all the sensitivity analyses, we took every tenth permutation (including the first one, [0,1,2,3,4]), for a total of 12 permutations. The order of permutations was as generated by the Python library call *itertools.permutations*. The resultant sequences were: [0,1,2,3,4], [0,2,4,1,3], [0,4,2,1,3], [1,2,0,3,4 , 1,3,4,0,2 , 2,0,3,1,4 , 2,3,0,1,4 , 2,4,3,0,1 , 3,1,2,0,4 , 3,4,0,1,2 , 4,0,3,1,2 , 4,2,1,0,3]. Here [0,1,2,3,4] means that synapse 0 fires first, then synapse 1, then synapse two and so on. In *Figure 6—figure supplement 1* we modified *Equation 3* slightly by computing *Asequential* as the mean of 5 runs with the same, sequential stimuli. This was done because the stochastic spiking and chemical calculations introduced considerable variability in the estimate of *Asequential*. In this figure the *mean(Atot)* term in *Equation 3* took this mean value of *Asequential*, and *Atot* for the other 11 permutations, as the set over which the mean was computed.

Chemical models are available in the supplementary material in tabular form and as GENESIS/kkit files, and will be hosted on http://doqcs.ncbs.res.in and ModelDB. Electrical models are available in supplementary material in tabular form and as MOOSE scripts. The morphology file is from Neuro-Morpho.org (*Ascoli et al., 2007*; *Dougherty et al., 2012*) and attached in the supplementary material as the file VHC-neuron.CNG.swc.

## Acknowledgements

I acknowledge funding support from NCBS/TIFR and the Department of Science and Technology grant SR/CSI/66/2013 under the Cognitive Science Research Initiative. Large simulations were carried out in the NCBS Supercomputing facility. I acknowledge Arvind Kumar and Sahil Moza for comments on the project.

## Additional information

### Competing interests

USB: Reviewing editor, *eLife*.

### Funding

| Funder | Grant reference number | Author |
|---|---|---|
| National Centre for Biological Sciences | Plan 4142 | Upinder Singh Bhalla |
| Department of Science and Technology, Ministry of Science and Technology | SR/CSI/66/2013 | Upinder Singh Bhalla |

The funders had no role in study design, data collection and interpretation, or the decision to submit the work for publication.

### Author contributions

USB, Conceptualization, Resources, Data curation, Software, Formal analysis, Supervision, Funding acquisition, Validation, Investigation, Visualization, Methodology, Writing—original draft, Project administration, Writing—review and editing

### Author ORCIDs

Upinder Singh Bhalla, http://orcid.org/0000-0003-1722-5188

## Additional files

### Supplementary files

• Source code 1. This set of source files is provided to illustrate generation of numerical portions of *Figure 2*, *Figure 5*, and *Figure 6*. The contents of the zip file include a README.txt, which has the file information and running instructions. In addition, it has script files for various parts of the figures, as well as model specification files for the chemical channel, and morphological parameters. Files include:

• Supplementary file 1. The supplementary data file specifies chemical reactions and parameters for the MAPK model. It also specifies channel equations and parameters for the full cell model used in *Figures 6* and *7*. Finally, it specifies the channel distributions in the full cell model.

### Major datasets

The following previously published dataset was used:

| Author(s) | Year | Dataset title | Dataset URL | Database, license, and accessibility information |
|---|---|---|---|---|
| Ascoli GA, Donohue DE, Halavi M | 2007 | NeuroMorpho.org | http://neuromorpho.org/neuron_info.jsp?neuron_name=VHC-neuron | Publicly available at NeuroMorpho.org (accession no: NMO_09573) |

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
