## [Decision Letter]

Thank you for submitting your article "Synaptic input sequence discrimination on behavioral time-scales mediated by reaction-diffusion chemistry in dendrites" for consideration by *eLife*. Your article has been favorably evaluated by Timothy Behrens (Senior Editor) and three reviewers, one of whom, Frances K Skinner (Reviewer #1), is a member of our Board of Reviewing Editors. The following individuals involved in review of your submission have agreed to reveal their identity: Paul Smolen (Reviewer #2) and Kim T Blackwell (Reviewer #3).

The reviewers have discussed the reviews with one another and the Reviewing Editor has drafted this decision to help you prepare a revised submission.

While all of the reviewers felt that this was interesting and significant work, there was a clear consensus that several aspects of the paper were unclear and thus subsequently confusing.

Overall, it was felt that the author needed to expand on his explanations and descriptions in several places. The specific points that need expansion, revision and further explanations in the paper are:

1) the 4 models

2) Q equation has issues

3) sensitivity of the results

Besides this, the reviewers have various issues in their comments (details that seem to need fixing) that should be responded to (see below).

*Reviewer #1:*

This is an interesting theoretical, computational study that considers sequence discrimination at the level of the single cell (rather than networks) including slower timescales (hundreds and milliseconds and slower). Reaction-diffusion modeling and detailed multi-scale models are used to show attributes of recognizing input in the correct order and over scrambled input and background noise, as well as showing an effect on firing.

While interesting, the manuscript could be more fully explained and expanded in several places to help the reader fully grasp the situation. Also, various details seemed a bit strange to me, and further discussion on some points seem warranted. Specifics are:

1) The author gives the context of place cells (Figure 1), but it would be helpful to explain this a bit further I think. That is, each neuronal ensemble sends a projection to a dendritic segment etc. Should this be considered as a particular postsynaptic cell? Should the network context be ignored, is this on multiple segments of a given cell, cells in a network, or does it matter in the context, etc.? I realize that the focus is on the single cell, and perhaps it doesn't really matter overall for the work, but some consideration/description of this would be helpful to the reader in setting the envisioned context. This would be further helpful in the Discussion when timescales are estimated and discussed.

2) For Figure 2, the author could expand on his 4 models and molecules A and B and Ca – that is, a schematic of the overall relationship would be helpful. Perhaps some sort of simple MAPK pathway (generic as for the 4 models) could be illustrated? The author says "The results were as expected" (subsection “Reaction systems select for distinct speeds and length-scales of sequential input”) in referring to Figure 7 – expand on why this is please. Figure 3 process not quite clear to me – left x-axis 'sequence score' is obtained how? The color on the right is Q, right? – please label.

It was not immediately obvious to me how a Q of -0.001 would be obtained for the sequence given – please explain and/or demonstrate/illustrate more fully.

2-6 μm spatial intervals given in text but in figures, the high Q values (color) are at tens of μm – I think that I am missing something or perhaps confused.

3) Multi-compartment model details are specified as taken from mostly from Traub et al. 1991, but the reference is not listed in the bibliography. Potassium reversal potential of -15 mV seems a bit depolarized, and I assume 60 mV for resting is a typo (-60 mV?). 1 ohm.m2 is 1000mS/cm2 for Rm, and seems rather high? The values in Figure 7 to see effect in firing are even larger by a lot which is perhaps a bit concerning?

4) Does it matter that there is no Ih in the model, which is well-known to be non-uniformly distributed in pyramidal dendrites. Would this be expected to affect the results?

5) Subsection “Sequence speed selectivity scales with reaction rates and diffusion constants” – "As expected, we found that as the rates were increased…". Why? As there was not a complete explanation for the 4 models, this 'expectation' suffers here too.

*Reviewer #2:*

This is an interesting paper dealing with an important topic in computational neuroscience, sequence discrimination. But some important points need to be addressed. I am asking for a little extra work – not an exhaustive analysis – in only one of my comments, comment 1, given that parameter sensitivity analysis is a common, substantial concern in all modeling studies.

1) Some parameter sensitivity analysis should be done for some of the key results. For example, in Figure 2, rightmost column, it appears that parameter values may be "tuned" to be close to bistability, so that stimuli can give relatively long, but eventually declining, response plateaus. Although, the numbers in the equations above do look pretty generic. Similarly, for Figure 2, and other figures that show strong sequence selectivity, how dependent is this selectivity on values of rate constants and other parameters? I'm not asking for an exhaustive analysis, but some degree of analysis and discussion of sensitivity should be given for a few of the key results. For example the sensitivity of the results of Figure 6, which presents selectivity for the most realistic model, should be discussed.

2) The selectivity criterion in Equation 3 cannot be correct as defined. The quantity Q, from Equation 2, is only related to the input, not the response, so Eq. 3 is also not related to the response. But of course selectivity, as plotted in the right column of Figure 3 and later, is related to the response. In Eq. 3, should Q be simply replaced by the average or maximum of the variable A, and in later cases should A be replaced by MAPK-P?

3) In the subsection “Sequence speed selectivity scales with reaction rates and diffusion constants” the author describes how by varying rates he can achieve good sequence selectivity over a broad range of time intervals, and in the Introduction it is noted that such time-invariance is a "desirable feature". But, for a single neuron or dendrite, the rate constants are presumably fixed, or at least do not vary in anticipation of stimulus time scales? So, I don't see how to get time invariance in a single neuron. So then, how does the author envision getting time invariance in a real neural network? Does he envision different neurons in the network will have different parameters so that they will respond to different time scales? These issues should be addressed in the Discussion.

4) For the channel modulations that underlie the simulations of Figure 7, it is never discussed whether these modulations would result from activation of the MAPK signaling pathway simulated in the rest of the paper. The channel modulations are imposed ad hoc. It would be good to connect these portions of the paper, if not by simulations of MAPK effects on channels, then at least by discussing to what extent some or all of these modulations are thought to be downstream of MAPK activation and are a plausible readout or result of the sequentially amplified activation of the MAPK pathway simulated previously.

5) In the Discussion, large numbers are given for how many sequence discriminations a neuron can perform per second and how many sequences a neuron can discriminate. But I am skeptical of these estimates because no discussion is given of what it means operationally for a neuron, a cell, to "discriminate" a sequence. It seems to me that a change in electrical firing rate or spike timing is needed to say a neuron has discriminated a sequence. Or, a change in synaptic plasticity. A transient biochemical response in a dendrite, just by itself, doesn't seem to me sufficient to constitute "discrimination" by the neuron. These caveats should be noted and discussed.

6) In the Methods, it is stated with regard to the simulations of Figure 7 that we ran the model for 1 second to let the cell settle, before modulating channels. But clarification is needed since 1 second is not enough to allow simulated biochemical pathways to come to equilibrium in the models. Especially in the realistic simulations of Figure 6, how much simulated time was allowed from when biochemical variables were initialized to when stimuli were given.

7) Figure 4 is a particular instance of the sensitivity issue in comment 1. Here a 20% change in stimulus amplitude leads to a very large change in the selectivity pattern. Is this sensitivity of pattern to stimulus amplitude typical for the models?

*Reviewer #3:*

The research addresses the important problem of pattern recognition by neurons. Specifically, what mechanisms allow a neuron to discriminate sequence a-b-c from c-b-a, where a, b, c are each a different synaptic input. The ability to discriminate temporal pattern is more difficult than discriminating which synaptic inputs are present. This issue has been addressed in specialized, direction selective cells, but the more general problem has not been investigated. A major strength of the manuscript is that the authors use both a simple, analytical model as well as biophysically and biochemically realistic neuron models to demonstrate principles underlying the results and to demonstrate the plausibility of neurons in tissue accomplishing temporal discrimination. Using the simple models, the author analyzes and presents the critical factors constraining sequence specificity. By using realistic model, the author demonstrates that sequence specificity can be implemented by neurons in tissue, since these simulations include background synaptic inputs, including theta modulated GABA input. For the most part the results are quite clear. There are only two aspects that need better exposition.

1) Though the difference in selectivity is apparent in Figure 3, it is not clear what exactly is being plotted. I understand using R^2, which gives values between 1 (i.e., good) -1 (i.e., bad) for Q if m=1, but I don't understand why add the term m into Q Equation 2. In addition, why does Equation 3 use Q as the selectivity measure. The first time I read this page, I thought that Q was the selectivity measure. Then, I realized that Q was just a convenient way to obtain a monotonic x axis value, and that the important information was in a graph of A versus Q. But then Equation 3 claims to plot a selectivity value which is a function of Q, not A. A more detailed illustration of how values in Figure 3 are obtained (one value from right side graph and one value from left side graph) would be quite helpful. Also, where does the temporal interval come into this equation?

2) The results in the subsection “Local dendritic channel modulations may influence cellular firing”, describing the effect of channel modulation need a bit more explanation. In particular, it is quite surprising that an increase in leak conductance would increase firing rate. Since leak channels are typically potassium channels, I would expect an increase in leak conductance to not only make the cell less sensitive to depolarizing current, but also hyperpolarize the branch a bit, unless Eleak is rather depolarized. An intuitive explanation for this result would be helpful. Also, I don't understand how these manipulations are related to sequence discrimination. Wouldn't these channel modulations change firing rate for all sequences? Or perhaps the neuron can better discriminate the correct from incorrect sequence?

---

## [Author Response]

*While all of the reviewers felt that this was interesting and significant work, there was a clear consensus that several aspects of the paper were unclear and thus subsequently confusing.*

*Overall, it was felt that the author needed to expand on his explanations and descriptions in several places. The specific points that need expansion, revision and further explanations in the paper are:*

*1) the 4 models*

*2) Q equation has issues*

*3) sensitivity of the results*

I thank the editor and reviewers for their thoughtful comments. I have addressed all the points brought up. Briefly,

1) I have added schematics and text to explain the structure and derivation of the 4 models.

2) I have fixed a regrettable mistake in Equation 3, where I had a different meaning for the Q symbol from Equation 2.

3) I have performed parameter sensitivity analyses for the abstract and detailed models, and presented these in supplementary figures.

In addition, I have gone through the other comments by the reviewers and made numerous revisions to address these.

*Besides this, the reviewers have various issues in their comments (details that seem to need fixing) that should be responded to (see below).*

*Reviewer #1:*

*[…] While interesting, the manuscript could be more fully explained and expanded in several places to help the reader fully grasp the situation. Also, various details seemed a bit strange to me, and further discussion on some points seem warranted. Specifics are:*

*1) The author gives the context of place cells (Figure 1), but it would be helpful to explain this a bit further I think. That is, each neuronal ensemble sends a projection to a dendritic segment etc. Should this be considered as a particular postsynaptic cell? Should the network context be ignored, is this on multiple segments of a given cell, cells in a network, or does it matter in the context, etc.? I realize that the focus is on the single cell, and perhaps it doesn't really matter overall for the work, but some consideration/description of this would be helpful to the reader in setting the envisioned context. This would be further helpful in the Discussion when timescales are estimated and discussed.*

Thank you for pointing this out. I have rewritten the introductory paragraph and added extra explanatory words in the figure legend to clarify these points.

“We assumed that a single neuron from each of these ensembles projects onto a given postsynaptic cell, which is the focus of our analysis. The projections are ordered such that they converge onto a succession of spines located on a short stretch of dendrite on this cell, in the same spatial and temporal order as the activation of the ensembles. We ignored all other network context.”

I have also updated the Discussion section:

“We envision a network configuration where at the input level, there are ensembles of neurons that are active in a defined sequence in some sensory, motor or other context. We focus our analysis on a specific postsynaptic neuron, which receives a projection from each of the preceding ensembles, such that each synapse is located within a few microns of the next.”

*2) For Figure 2, the author could expand on his 4 models and molecules A and B and Ca – that is, a schematic of the overall relationship would be helpful. Perhaps some sort of simple MAPK pathway (generic as for the 4 models) could be illustrated?*

In Figure 2 have added a new row of panels (B) with schematics for the four models. I have updated the figure legend to explain these schematics.

*The author says "The results were as expected" (subsection “Reaction systems select for distinct speeds and length-scales of sequential input”) in referring to Figure 7 – expand on why this is please.*

I have expanded on this as follows: “If the chemical system is highly selective, we expect that only one or two points in the scatter plot should have a high value of *Atot*, and these should be the points with Q around +1 or -1. […] Each of these outcomes had been observed in Figure 2 for just the sequential and a single scrambled stimulus, and in Figure 3 we found that it generalized to the entire set of sequence permutations.”

*Figure 3 process not quite clear to me – left x-axis 'sequence score' is obtained how? The color on the right is Q, right? – please label.*

I apologize for lack of clarity around Equation 3 and this figure. The left x axis is Q and the color on the right is the selectivity, from Equation 3. This is now corrected on the figure and in the text.

*It was not immediately obvious to me how a Q of -0.001 would be obtained for the sequence given – please explain and/or demonstrate/illustrate more fully.*

I have introduced new panels A and B in Figure 3 to illustrate the computation of Q.

*2-6 μm spatial intervals given in text but in figures, the high Q values (color) are at tens of μm – I think that I am missing something or perhaps confused.*

This was indeed confusing, as a result of my using the total activated length rather than the spatial intervals between synapses, for just this figure. I have corrected it in the new Figure 3 and Figure 4.

*3) Multi-compartment model details are specified as taken from mostly from Traub et al. 1991, but the reference is not listed in the bibliography.*

Fixed.

*Potassium reversal potential of -15 mV seems a bit depolarized, and I assume 60 mV for resting is a typo (-60 mV?).*

Reversal potential is as per Traub 1991, but I agree it is a bit depolarized. I didn’t want to get into reconfigurations of the channel properties for the current study so I left it as is.

Corrected resting potential to -60 mV.

*1 ohm.m2 is 1000mS/cm2 for Rm, and seems rather high?*

This is 10,000 ohm.cm^2, or 0.1 mS/cm^2, which is what Traub used. I understand that this is a fairly typical value.

*The values in Figure 7 to see effect in firing are even larger by a lot which is perhaps a bit concerning?*

This is indeed an important point. In the revision I have strengthened the point that the required modulations are large. From the Discussion:

“While the amount of modulation required for an effect is quite large in some cases, we stress that these were proof-of-principle calculations. […] Similar small length-scale events have been shown to affect cellular firing in striatal neurons (Plotkin et al., 2011), suggesting that this outcome of sequence selection may also be plausible.”

*4) Does it matter that there is no Ih in the model, which is well-known to be non-uniformly distributed in pyramidal dendrites. Would this be expected to affect the results?*

I don’t think the absence of Ih affects the results. For the purposes of sequence recognition, the biggest impact would be if Ca^2+^ dynamics were to be affected, since Ca^2+^ is the input to the signaling pathways. This may indeed occur from Ih regulation of local excitability. However, in the revision, I have explored parameter sensitivity to four Ca^2+^ related parameters (Figure 6—figure supplement 1). In particular, LCa density could be varied from 0 to 10 S/m^2 without much effect on the selectivity. Given that even a Ca^2+^ channel isn’t having much effect, I feel that the omission of Ih is unlikely to change the core results.

*5) Subsection “Sequence speed selectivity scales with reaction rates and diffusion constants” – "As expected, we found that as the rates were increased…". Why? As there was not a complete explanation for the 4 models, this 'expectation' suffers here too.*

I have added a couple of lines to explain this. Additionally, as discussed above, the four models are explained in much more detail and with new figure panels.

“As expected, scaling all the chemical and diffusion rates also scaled the sequence speed to which the network had the strongest response. As the rates were increased, the best tuning was at shorter time intervals but roughly the same spatial intervals.”

*Reviewer #2:*

*This is an interesting paper dealing with an important topic in computational neuroscience, sequence discrimination. But some important points need to be addressed. I am asking for a little extra work – not an exhaustive analysis – in only one of my comments, comment 1, given that parameter sensitivity analysis is a common, substantial concern in all modeling studies.*

Thank you. I have now done some parameter sensitivity analysis as indicated below.

*1) Some parameter sensitivity analysis should be done for some of the key results. For example, in Figure 2, rightmost column, it appears that parameter values may be "tuned" to be close to bistability, so that stimuli can give relatively long, but eventually declining, response plateaus.*

The ‘switch’ model in the rightmost column is in fact a one-shot, that is, a switch with a delayed turnoff. This is now shown in the schematic in Figure 2, and also in the new text explaining how the abstract models were developed:

“To the bistable switch system we added inhibitory feedback to restore the switch to baseline after a delay.”

*Although, the numbers in the equations above do look pretty generic. Similarly, for Figure 2, and other figures that show strong sequence selectivity, how dependent is this selectivity on values of rate constants and other parameters? I'm not asking for an exhaustive analysis, but some degree of analysis and discussion of sensitivity should be given for a few of the key results. For example the sensitivity of the results of Figure 6, which presents selectivity for the most realistic model, should be discussed.*

Thank you for bringing this up, it is a matter that is important and is now systematically addressed. These are in Figure 4—figure supplement 1 and 2 and are described in the following new paragraph:

“To better understand the range of sequence selectivity, we varied each of the parameters of the model one at a time with respect to the reference model. […] These models were intended to be illustrative and to explore the properties of sequence-selective systems. Hence we did not optimize the models for robustness.”

I have also done a smaller parameter sweep for the full cell model, which is not exhaustive because there are a large number of parameters and it would be slow to do a sweep for all parameters (~12 CPU hours each run * 12 sequences * 4 technical repeats * (6 to 8) values per parameter):

“We performed a limited parameter sensitivity analysis for the detailed cell model, focusing on biophysical and biochemical parameters involved in triggering or responding to Ca^2+^. […] While this list is not exhaustive, it does suggest that the sequence selectivity of the detailed model is fairly robust, and somewhat more so than the abstract models.”

*2) The selectivity criterion in Equation 3 cannot be correct as defined. The quantity Q, from Equation 2, is only related to the input, not the response, so Eq. 3 is also not related to the response. But of course selectivity, as plotted in the right column of Figure 3 and later, is related to the response. In Eq. 3, should Q be simply replaced by the average or maximum of the variable A, and in later cases should A be replaced by MAPK-P?*

I am sorry about this mistake. I had mistakenly used Q to refer to the variable now called Atot in the text. The reviewer correctly identifies A as the variable in question, and MAPK-P in later figures.

“*Selectivity = (Asequential – mean(Atot))/max(Atot)* Equation 3”

*3) In the subsection “Sequence speed selectivity scales with reaction rates and diffusion constants” the author describes how by varying rates he can achieve good sequence selectivity over a broad range of time intervals, and in the Introduction it is noted that such time-invariance is a "desirable feature". But, for a single neuron or dendrite, the rate constants are presumably fixed, or at least do not vary in anticipation of stimulus time scales? So, I don't see how to get time invariance in a single neuron. So then, how does the author envision getting time invariance in a real neural network? Does he envision different neurons in the network will have different parameters so that they will respond to different time scales? These issues should be addressed in the Discussion.*

In the Discussion I have added the following paragraph under the heading “Single-cell sequence recognition may act on a continuum of timescales”:

“Individual chemical implementations of sequence recognition in our study operate over a range of about 2-5 microns and 1.5-4 seconds (Figure 4, Figure 6). […] It is interesting to speculate that there may be useful computational implications of having different cells ‘tuned’ to different sequence speeds.”

*4) For the channel modulations that underlie the simulations of Figure 7, it is never discussed whether these modulations would result from activation of the MAPK signaling pathway simulated in the rest of the paper. The channel modulations are imposed ad hoc. It would be good to connect these portions of the paper, if not by simulations of MAPK effects on channels, then at least by discussing to what extent some or all of these modulations are thought to be downstream of MAPK activation and are a plausible readout or result of the sequentially amplified activation of the MAPK pathway simulated previously.*

I thank the reviewer for this useful suggestion to improve the linkage between the chemical events and the electrical events. I have added text to explain how the simulations were run and how MAPK or other signaling may be upstream of these channel modulations.

“We assumed that MAPK activity, or other signaling events downstream of sequence selective chemistry, could perform channel modulation through phosphorylation or triggering of channel insertion. […] Instead we ran just the electrical cell model for an initial settling period of 1 second, and then used the simulation script to modify the selected channel conductance to represent its modulation. Following this we ran the model for another 3 seconds in each case.”

*5) In the Discussion, large numbers are given for how many sequence discriminations a neuron can perform per second and how many sequences a neuron can discriminate. But I am skeptical of these estimates because no discussion is given of what it means operationally for a neuron, a cell, to "discriminate" a sequence. It seems to me that a change in electrical firing rate or spike timing is needed to say a neuron has discriminated a sequence. Or, a change in synaptic plasticity. A transient biochemical response in a dendrite, just by itself, doesn't seem to me sufficient to constitute "discrimination" by the neuron. These caveats should be noted and discussed.*

I agree that this discussion should have been related more clearly to the analysis in Figure 7, and to other possible downstream effects of sequence recognition. In order to address this, I have moved the Discussion section “Plasticity is a likely target…” to the first part of the Discussion, and I have renamed and substantially expanded this section to indicate the kinds of outcome one may get from sequence recognition chemistry. It is now headed “Firing change, synaptic plasticity and morphological change may result from biochemical sequence recognition”. This leads into the discussion on “Sequence recognition confers substantial computational capabilities on single cells”. I have further added the following lines to put it in context:

“For simplicity, we consider computation in terms of immediate firing rate changes (discussed above), though plasticity and morphological change are also computational outcomes.”

*6) In the Methods, it is stated with regard to the simulations of Figure 7 that we ran the model for 1 second to let the cell settle, before modulating channels. But clarification is needed since 1 second is not enough to allow simulated biochemical pathways to come to equilibrium in the models. Especially in the realistic simulations of Figure 6, how much simulated time was allowed from when biochemical variables were initialized to when stimuli were given.*

I agree that this was not explained sufficiently. In the main text I have added a paragraph to explain the procedure:

“We did not simulate these modulation events explicitly since the computations were very lengthy and we wished to perform many repeats to estimate the distribution of the firing rates following channel modulation. Instead we ran just the electrical cell model for an initial settling period of 1 second, and then used the simulation script to modify the selected channel conductance to represent its modulation. Following this we ran the model for another 3 seconds in each case.”

*7) Figure 4 is a particular instance of the sensitivity issue in comment 1. Here a 20% change in stimulus amplitude leads to a very large change in the selectivity pattern. Is this sensitivity of pattern to stimulus amplitude typical for the models?*

Please see response to comment 1. Yes, several parameters are quite sensitive.

“We found that selectivity was robust to some parameters, such as diffusion constants, but fragile (range ~20%) with respect to some rate constants and stimulus amplitude. These models were intended to be illustrative and to explore the properties of sequence-selective systems. Hence we did not optimize the models for robustness.”

*Reviewer #3:*

*[…] 1) Though the difference in selectivity is apparent in Figure 3, it is not clear what exactly is being plotted. I understand using R^2, which gives values between 1 (i.e., good) -1 (i.e., bad) for Q if m=1, but I don't understand why add the term m into Q Equation 2.*

I apologize for several points of confusion in this section. To address this, I have added two plots to Figure 3 to illustrate how the metric Q works. I have also added some text:

“*R^[2]^* provides a measure of linearity of the sequence, and *m* assigns it a magnitude and sign. With this measure, the sequence [0,1,2,3,4] has Q = 1; [4,3,2,1,0] has Q = -1, and [4,0,2,1,3] has Q = -0.001. Examples of regression plots used to generate Q for different sequences are presented in Figure 3.”

*In addition, why does Equation 3 use Q as the selectivity measure. The first time I read this page, I thought that Q was the selectivity measure. Then, I realized that Q was just a convenient way to obtain a monotonic x axis value, and that the important information was in a graph of A versus Q. But then Equation 3 claims to plot a selectivity value which is a function of Q, not A.*

This was an error on my part, and I had used the term Q incorrectly in Equation 3. It is now fixed:

*“Selectivity = (Asequential – mean(Atot))/max(Atot)* Equation 3”

Provided the system responds more strongly to sequential input than to any other order, *selectivity* should be in the range from zero (unselective) to one (highly selective). *Asequential* is *Atot* for the sequential stimulus, *mean(Atot)* is the mean of *Atot* over all permutations of the stimulus order, and *max(Atot)* is the maximum value of *Atot* for all permutations of stimulus order.”

*A more detailed illustration of how values in Figure 3 are obtained (one value from right side graph and one value from left side graph) would be quite helpful.*

I now indicate (using asterisks), which point on the right hand heatmap plot comes from the left hand graph, in Figure 3 also specify (using crosses) which of the points on the left hand panel in Figure 3 comes from the examples in Figure 2.

*Also, where does the temporal interval come into this equation?*

For Equation 2 I state: “We plotted the ordinal position of the input against the ordinal value of its arrival time.” So, temporal interval is factored out.

In Equation 3, the temporal and spatial intervals must be supplied to the simulation as control parameters. I have added the line:

“The time and space intervals were specified as parameters for each simulation run.”

*2) The results in the subsection “Local dendritic channel modulations may influence cellular firing”, describing the effect of channel modulation need a bit more explanation. In particular, it is quite surprising that an increase in leak conductance would increase firing rate. Since leak channels are typically potassium channels, I would expect an increase in leak conductance to not only make the cell less sensitive to depolarizing current, but also hyperpolarize the branch a bit, unless Eleak is rather depolarized. An intuitive explanation for this result would be helpful.*

Thank you for pointing out this source of confusion. I had implemented the leak conductance as a nonselective cation current leak, which reverses close to 0 mV. I now state this explicitly in the text and figure legend. In preliminary calculations I had also played with modulating Ih, the HCN channel, but since its voltage-dependent effects were not incorporated into the cell model I didn’t pursue this.

*Also, I don't understand how these manipulations are related to sequence discrimination. Wouldn't these channel modulations change firing rate for all sequences? Or perhaps the neuron can better discriminate the correct from incorrect sequence?*

This is an important point that other reviewers have also brought up. I now have new paragraphs in the main text and in the Discussion to explain the intended link between the chemical events of sequence discrimination, and the channel modulation. These are detailed above.